# LABEL PRIVACY SOURCE CODING IN VERTICAL FEDERATED LEARNING

## ABSTRACT

We study label privacy protection in vertical federated learning (VFL). VFL enables an active party who possesses labeled data to improve model performance (utility) by collaborating with passive parties who have auxiliary features. Recently, there has been a growing concern for protecting label privacy against semi-honest passive parties who may surreptitiously deduce private labels from the output of their bottom models. However, existing studies do not remove the prior label information in the active party's features from labels in an offline phase, thus leaking unnecessary label privacy to passive parties. In contrast to existing methods that focus on training-phase perturbation, we propose a novel offline-phase data cleansing approach to protect label privacy without compromising utility. Specifically, we first formulate a Label Privacy Source Coding (LPSC) problem to remove the redundant label information in the active party's features from labels, by assigning each sample a new weight and label (i.e., residual) for federated training. We give a privacy guarantee and theoretically prove that *gradient boosting* efficiently optimizes the LPSC problem. Therefore, we propose the *Vertical Federated Gradient Boosting* (VFGBoost) framework to address the LPSC problem. Moreover, given that LPSC only provides upper-bounded privacy enhancement, VFGBoost further enables a flexible privacy-utility trade-off by incorporating *adversarial training* during federated training. Experimental results on four real-world datasets substantiate the efficacy of LPSC and the superiority of our VFGBoost framework.

## 1 INTRODUCTION

Vertical federated learning (VFL) Yang et al. (2019) enables global model construction among organizations with datasets sharing overlapping sample spaces but differing feature spaces. Fig. 1(a) presents an overview of the multi-party VFL problem, where an active party possesses labeled data and has aligned samples with passive parties that own auxiliary features. The primary goal of VFL is to build a well-performed federated model in a privacy-preserving and efficient manner.

Recently, label privacy protection has attracted increasing attention in VFL studies. Existing studies in VFL label privacy protection Li et al. (2022); Fu et al. (2022); Sun et al. (2022) rely on a model-splitting paradigm, as shown in Fig. 1(b), in which a DNN model is divided into a top model and bottom models to protect label privacy and feature privacy, respectively. They protect label privacy by training a complex-yet-deterministic top model with various perturbation techniques. However, when a passive party steals the deterministic top model (e.g., via model completion attack Fu et al. (2022)), the worst-case label privacy leakage risk occurs and approximates the federated model performance, which is the utility Sun et al. (2022). More concerning, the high-dimensional forward embeddings adopted for label protection exacerbate the feature privacy leakage to the active party Jin et al. (2021); Ye et al. (2022). The fundamental cause of this dilemma is that existing studies Li et al. (2022); Fu et al. (2022); Sun et al. (2022) directly optimize the forward embeddings for *label prediction*, making forward embeddings highly correlated with and informative about private labels.

Our key insight is that label privacy protection in VFL should be decoupled into two independent tasks: 1) *offline-phase cleansing*, which enhances privacy without compromising utility by removing the redundant label information from labels, and 2) *training-phase perturbation*, which further balances privacy-utility trade-off via inadequately learning from perturbed labels or gradients.

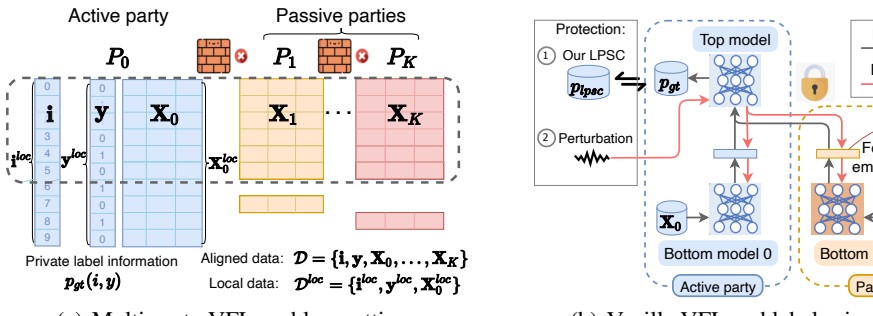

(a) Multi-party VFL problem setting     (b) Vanilla VFL and label privacy threat

Figure 1: (a) The multi-party VFL problem setting. (b) Vanilla VFL trains model with *uniformly-weighted original labels* $p_{gt}(i, y)$. A semi-honest passive party attacks label privacy from the forward embedding. Our LPSC replaces $p_{gt}(i, y)$ with optimized *re-weighted residuals* $p_{lpsc}(i, y)$.

As a remedy to the aforementioned loophole, we formulate a Label Privacy Source Coding (LPSC) problem to encode *minimum-sufficient* label privacy in an offline phase. The idea is to remove the label information present in the active party's local features, which is redundant for VFL, from the ground-truth label. By doing so, the risk of label leakage from forward embeddings is significantly eliminated, without sacrificing utility. We theoretically analyze the privacy guarantee of LPSC.

LPSC is a constrained optimization problem of two mutual information. We prove that *gradient boosting* Freund & Schapire (1997) is a simple and efficient approach to optimize the LPSC problem. Specifically, gradient boosting converts the uniformly-weighted original labels to *re-weighted residuals* of the active party's local predictions, thus eliminating the redundant label privacy.

Therefore, we propose Vertical Federated Gradient Boosting (VFGBoost) to shift the federated learning target from the uniformly-weighted original labels $p_{gt}(i, y)$ to the LPSC-encoded re-weighted residuals $p_{lpsc}(i, y)$, which encodes *minimum-sufficient* label privacy for federated training. Our proposed VFGBoost follows the aforementioned two-phase paradigm: In the offline LPSC phase (Fig. 2, phase 1), the active party trains a local model on its local data and computes the LPSC-encoded re-weighted residuals $p_{lpsc}(i, y)$ via gradient boosting as the learning target for VFL. Subsequently, in the federated training phase (Fig. 2, phase 2), the passive parties train a federated model to fit the re-weighted residuals. Hence, the federated prediction is the weighted sum of the active party's local prediction and the federated predicted residual.

Crucially, the inherent label privacy enhancement of LPSC is *upper-bounded* by the label information present in the active party's local features, potentially falling short in practical scenarios. To circumvent this, perturbation methods can be subsequently employed to enhance label privacy with a consequent reduction in utility. Specifically, VFGBoost utilizes *adversarial training* through a max-min optimization, in the federated training phase (Fig. 2, phase 2). The active party trains adversarial top models by simulating adversaries to attack labels, while also updating the passive parties' bottom models to thwart the attack. Consequently, VFGBoost consists of a *utility objective* that learns to fit the LPSC-encoded label privacy (re-weighted residuals) $p_{lpsc}(i, y)$, as well as an adversarial *privacy objective* that further protects ground-truth label privacy $p_{gt}(i, y)$. We jointly optimize both objectives, utilizing a hyperparameter to enable flexible balancing of the privacy-utility trade-off. Moreover, VFGBoost is model-agnostic and allows any gradient-based model.

Our comprehensive experiments conducted on four real-world datasets in the realms of recommendation and healthcare demonstrate that the LPSC can enhance label privacy without compromising utility, and the proposed VFGBoost framework achieves a superior privacy-utility trade-off compared to seven baseline methods. In summary, our contributions are as follows:

- We decouple label privacy protection in VFL into two independent tasks: *offline-phase cleansing* to inherently enhance privacy without compromising utility, and *training-phase perturbation* for nuanced privacy-utility trade-offs.

- We formulate a Label Privacy Source Coding (LPSC) problem with privacy guarantee to encode *minimum-sufficient* label information for offline-phase cleansing.

- We further propose VFGBoost that utilizes *gradient boosting* to optimize LPSC and incorporates *adversarial training* to enable additional privacy enhancement.

- We perform extensive experiments on four real-world datasets to demonstrate the efficacy of LPSC and the superiority of our proposed VFGBoost framework.

## 2 RELATED WORK

**Label Privacy Protection in VFL.** Existing label privacy protection techniques in VFL mainly include cryptographic methods and perturbation methods. **Cryptographic methods** Fu et al. (2021); Cheng et al. (2021); Ren et al. (2022) incur significant overheads in computation and communication, which is typically unbearable in practice. Therefore, they are *not* investigated and compared in this work. **Perturbation methods** introduce noise to labels or gradients to update the passive parties' models. For instance, Li et al. Li et al. (2022) employ adapted Gaussian noise to perturb the gradients to defend against label attacks. Sun et al. Sun et al. (2022) minimize the distance correlation between the forward embedding and the label to defend against the spectral attack Tran et al. (2018). Ghazi et al. Ghazi et al. (2021) leverage randomized responses to use randomly flipped labels for computing gradients. Yang et al. Yang et al. (2022) apply differential privacy Dwork et al. (2006) to a gradient perturbation-based split learning framework. Overall, due to the forward embeddings in existing works being optimized for label prediction Li et al. (2022); Zou et al. (2022); Sun et al. (2022), the worst-case label privacy leakage approximates the VFL utility, which is unacceptable.

**Mutual Information for Privacy Protection.** MID Zou et al. (2023) uses mutual information (MI) regularization to minimize the entropy of forward embedding during federated training. It adopts a VAE-based MI estimator Alemi et al. (2016) to gauge MI between the embedding and label. Such explicit MI estimation Alemi et al. (2016); Belghazi et al. (2018), however, is resource-intensive and needs Gaussian noise, reducing utility. Conversely, our LPSC employs gradient boosting to enhance privacy efficiently without added noise.

**Privacy Protection via Offline Pre-processing.** Recently, InstaHide Huang et al. (2020) and Fed-Pass Gu et al. (2023) are proposed to pre-process features to safeguard feature privacy by merging training samples or adding noise. Nevertheless, to our best knowledge, there are no existing pre-processing approaches designed for label privacy protection.

## 3 PROBLEM FORMULATION

**Vertical Federated Learning Setting.** In a typical VFL setting, as shown in Fig. 1(a), the aligned training data $\mathcal{D}$ has sample identifiers (IDs) $i$ and labels $y$. The feature matrix $X = [X_0, X_1, \ldots, X_K]$ is vertically partitioned among $K + 1$ parties by feature. An active party $P_0$ has labeled local features $\{i, X_0, y\}$. Meanwhile, $K$ passive parties $\{P_k\}_{k=1}^K$ only have auxiliary features $\{i, X_k\}_{k=1}^K$. Moreover, the active party has local data $\mathcal{D}^{loc} = \{i^{loc}, X_0^{loc}, y^{loc}\}$, which additionally includes unaligned labeled samples. In VFL, the active party aims to leverage the auxiliary features from passive parties to train a federated model while protecting privacy. For simplicity, **we use sample ID $i$ to represent $P_k$'s features $x_{k,i}$ in functions** (e.g., $h_{\psi_k}(i)$ denotes $h_{\psi_k}(x_{k,i})$). A summary of notations and their corresponding descriptions is provided in the Appendix A.

Table 1: Threat model. (See details in Appendix B.)

| Threat model | Adversary | Attack objective | Attack method | Adversary's capability |
|---|---|---|---|---|
| Semi-honest | Passive parties | $\min \mathbb{R}_{p_{gt}(i,y)}$ | Norm, Spectral, PMC | A few labeled samples |
| | Active party | $\min \mathbb{R}_{p_{gt}(i,x)}$ | Model inversion | A few samples with features |

**Threat Model.** We focus on privacy leakage stemming from the forward embedding of passive parties' bottom models. We assume that both active and passive parties are *semi-honest* and non-colluding, meaning that they follow the training protocol but attempt to extract private information. To attack label privacy, an adversarial passive party $P_k$ minimizes the following error $\mathbb{R}_{p_{gt}(i,y)}$ against the ground-truth ID-label joint distribution $p_{gt}(i, y)$, given $D_{\mathrm{KL}}(\cdot||\cdot)$ as KL-divergence:

$$\min_{A \in \mathbb{A}} \mathbb{R}_{p_{gt}(i,y)}(A \circ h_{\psi_k}) = \min_{A \in \mathbb{A}} \mathbb{E}_{i \sim p_{gt}(i)}[D_{\mathrm{KL}}(p_{gt}(y|i)||A(h_{\psi_k}(i)))], \tag{1}$$

where $A \in \mathbb{A}$ is *any* attack function that infers the raw label from $P_k$'s forward embedding $h_{\psi_k}(i)$. Every attack method corresponds to a different attack function $A(\cdot)$. We assume the adversary $P_k$ has *no* prior knowledge of $p_{gt}(i, y)$, which is *different from differential privacy's assumption*.

The feature attack objective can be formulated similarly. As shown in Table 1, in our threat model, a passive party adversary uses norm attack Li et al. (2022), spectral attack Tran et al. (2018), or passive model completion (PMC) attack Fu et al. (2022) to build the attack function $A(\cdot)$. Similarly, an active adversary uses model inversion attack He et al. (2019) to attack features.

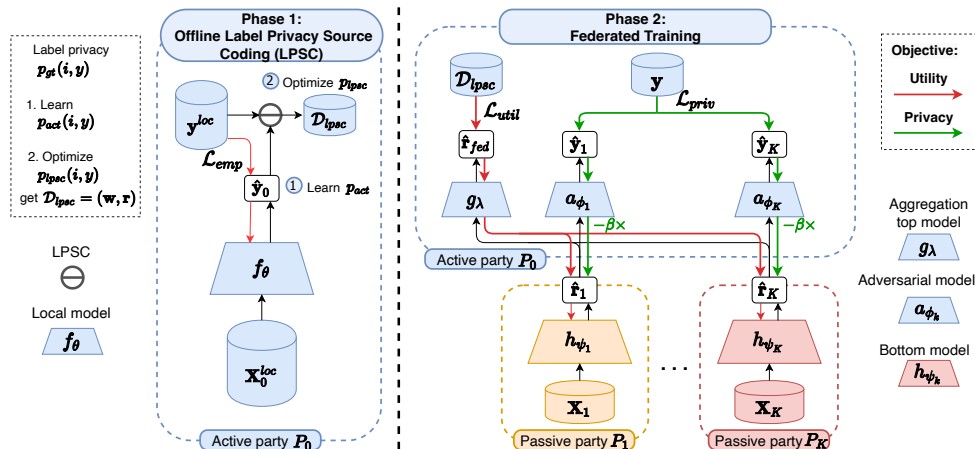

Figure 2: The VFGBoost framework. Left: the offline Label Privacy Source Coding (LPSC) phase (Section 4.2). Right: the federated training phase (Section 4.3).

**Privacy Definition.** According to Equation 1, the adversary's objective is to minimize the *expected* estimation error of the ground-truth conditional distribution $p_{gt}(y|i)$. Therefore, the *private label information is and only is* the ID-label joint distribution $p_{gt}(i, y)$, see details in Appendix B.2.

Our goal is to design an offline-phase privacy mechanism $\mathcal{M}$ for the active party $P_0$ that outputs a new joint distribution $p_{\mathcal{M}}(i, y) = \mathcal{M}(p_{gt}(i, y, X_0))$. Subsequently, the bottom model $h_{\psi_k}$ is trained to fit $p_{\mathcal{M}}(i, y)$ during federated training. Therefore, the optimal attack error given $p_{\mathcal{M}}(i, y)$ is:

$$\mathbb{R}_{p_{gt}(i,y)}(p_{\mathcal{M}}(i, y)) = \min_{A \in \mathbb{A}} \mathbb{E}_{i \sim p_{gt}(i)}[D_{\mathrm{KL}}(p_{gt}(y|i)||A(h_{\psi_k^*}(i)))] \quad \text{(Attack label)}$$

$$\text{where} \quad \psi_k^* = \arg\min_{\psi_k} \mathbb{E}_{i \sim p_{\mathcal{M}}(i)}[D_{\mathrm{KL}}(p_{\mathcal{M}}(y|i)||g(h_{\psi_k}(i)))] \quad \text{(Train bottom model)},$$

where $g(\cdot)$ is $P_0$'s top model trained to map $h_{\psi_k}(i)$ to the new label $p_{\mathcal{M}}(y|i)$. We notice that, $\max_{p_{\mathcal{M}}(i,y)} \mathbb{R}_{p_{gt}(i,y)}(p_{\mathcal{M}}(i, y)) \iff \min_{p_{\mathcal{M}}(i,y)} I(p_{gt}(i, y); p_{\mathcal{M}}(i, y))$. Thereby, our threat model coincides with $\epsilon$-*mutual information privacy ($\epsilon$-MIP)*, see more details in Appendix B.

## 4    PROPOSED APPROACH

In this section, we introduce our two-phase VFGBoost framework that consists of an offline LPSC phase and a federated training phase, as depicted in Fig. 2. Specifically, we first formulate the LPSC problem (Section 4.1) that encodes minimum-sufficient label privacy by removing redundant label information from the active party's features. Then, we prove that gradient boosting can efficiently optimize the LPSC problem (Section 4.2). Thereby, we proposed our VFGBoost framework that leverages gradient boosting to tackle LPSC. Moreover, to flexibly balance the privacy-utility trade-off, VFGBoost incorporates *adversarial training* in the federated training phase (Section 4.3). The active party mimics adversarial passive parties to attack labels and, in turn, solves a max-min optimization problem to protect labels.

## 4.1    LABEL PRIVACY SOURCE CODING PROBLEM

In the offline phase (Fig. 2, phase 1), we aim to encode *minimum-sufficient* label privacy from the ground-truth label privacy $p_{gt}(i, y)$, by removing the redundant label information $p_{act}(i, y)$ in the active party's local features $X_0$, as demonstrated in Fig. 3. To do so, we formally define a label privacy source coding problem as follows:

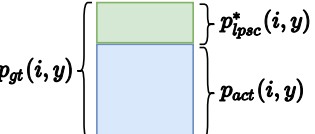

Figure 3: Schematic graph of LPSC. $p_{lpsc}^*$ is optimal $p_{lpsc}$.

**Problem 1** (Label Privacy Source Coding)**.** *Given ground-truth label privacy $p_{gt}(i, y)$ and the active party $P_0$'s learned label privacy $p_{act}(i, y)$ from its features $X_0$, the label privacy source coding problem is to optimize a new ID-label joint distribution $p_{lpsc}(i, y)$ as follows:*

$$\max_{p_{lpsc}(i,y)} I(p_{gt}(i, y); p_{lpsc}(i, y)) \qquad \text{(Sufficient)} \qquad (2)$$

$$s.t. \ I(p_{act}(i, y); p_{lpsc}(i, y)) = 0 \qquad \text{(Minimum)},$$

*where $I(\cdot; \cdot)$ denotes mutual information.*

The optimized ID-label joint distribution $p_{lpsc}(i, y)$ assigns each sample a new weight through the marginal $p_{lpsc}(i)$ and/or label through the conditional $p_{lpsc}(y|i)$. We will show that gradient boosting, which is detailed in the Appendix C.1, efficiently solves the LPSC problem. The privacy leakage inherent in LPSC is rigorously bounded as described in the following theorem:

**Theorem 1** (Privacy Guarantee). *LPSC satisfies $\epsilon$-mutual information privacy ($\epsilon$-MIP). The privacy leakage is bounded by $\epsilon = H(p_{gt}(i, y)|p_{act}(i, y))$, the conditional entropy of the ground-truth label distribution $p_{gt}(i, y)$ given the active party's label distribution $p_{act}(i, y)$. Formally,*

$$I(p_{gt}(i, y); p_{lpsc}^*(i, y)) \le \epsilon \text{ bits,}$$

*where $p_{lpsc}^*(i, y)$ represents the optimal solution of Equation 2 in the LPSC problem.*

**Proof.** The proof of Theorem 1 is provided in the Appendix B.3. The intuition behind Theorem 1 is that privacy leakage in LPSC is inversely related to the amount of label information the active party can infer from its local features.

## 4.2 GRADIENT BOOSTING SOLVES LPSC PROBLEM

A recent insight of mutual information (MI) regularization for privacy protection Zou et al. (2023) relies on a notion of MI neural estimation Alemi et al. (2016); Belghazi et al. (2018), which explicitly estimates MI via Gaussian noise. However, explicit MI estimation is inefficient, and the introduced noises hinder model utility Belghazi et al. (2018). In contrast, we prove that gradient boosting is a simple and efficient approach to solve the LPSC problem.

To solve problem 1, 1) the active party $P_0$ first learns the label privacy $p_{act}(i, y)$ present in its features $\boldsymbol{X}_0^{loc}$. 2) Then, active party $P_0$ optimizes the joint distribution $p_{lpsc}(i, y)$ by solving Eq. 2. We elaborate on each step as follows:

**(1) Learning $p_{act}(i, y)$.** To learn $p_{act}(i, y)$, which is the label privacy present in local features $\boldsymbol{X}_0$, the active party $P_0$ only needs to learn the conditional $p_{act}(y|i)$ as the marginal $p_{act}(i) = p_{gt}(i) \sim U$ is uniform. To do so, $P_0$ trains model $f_\theta$ on its local data $\mathcal{D}^{loc}$ indexed by $\boldsymbol{i}^{loc}$ as follows:

$$\theta^* = \arg\min_\theta \sum_{i \in \boldsymbol{i}^{loc}} \frac{1}{|\boldsymbol{i}^{loc}|} \mathcal{L}_{emp}(y_i, f_\theta(i)), \tag{3}$$

where $\mathcal{L}_{emp}$ denotes the empirical loss. $f_\theta(i)$ denotes $f_\theta(\boldsymbol{x}_{0,i})$ for simplicity and models the conditional label distribution $p_{act}(y|i)$. Consequently, the active party learns $p_{act}(i, y) = p_{gt}(i) \cdot p_{act}(y|i)$.

**(2) Optimizing $p_{lpsc}(i, y)$.** We point out that the *gradient boosting algorithm optimizes the LPSC problem* by taking AdaBoost Freund & Schapire (1997) as an example. As shown in Theorem 2 and Theorem 3, we prove that the AdaBoost algorithm optimizes the LPSC problem by minimizing the KL-divergence between $p_{lpsc}(i)$ and the uniform distribution $U$ (Eq. 4), while fixing the conditional distribution $p_{lpsc}(y|i)$ as the ground truth $p_{gt}(y|i)$.

**Theorem 2.** *Assuming fixed conditional distribution $p_{lpsc}(y|i) = p_{gt}(y|i)$ and let $U$ denote uniform distribution, the LPSC problem 1 can be reduced to:*

$$\min_{p_{lpsc}(i)} D_{\text{KL}}(p_{lpsc}(i) \parallel U) \quad s.t. \sum_{i \in \boldsymbol{i}} p_{lpsc}(i) y_i f_\theta(i) = 0, \tag{4}$$

*where $i \in \boldsymbol{i}$ is the sample index of aligned training data with IDs $\boldsymbol{i}$.*

**Proof.** The proof of Theorem 2 is provided in the Appendix D.1. Theorem 2 reduces the LPSC problem to a convex optimization problem Eq. 4, which can be solved via Lagrangian. It projects the ground-truth label privacy $p_{gt}(i, y)$ to an information plane that is orthogonal to $p_{act}(i, y)$, thus eliminating the redundant label information in active party's features $\boldsymbol{X}_0$.

**Theorem 3.** *Schapire & Freund (2013) The solution of the convex optimization problem Eq. 4 is equivalent to AdaBoost Freund & Schapire (1997):*

$$p_{lpsc}(i) = \frac{e^{-\alpha y_i f_\theta(i)}}{\sum_{i \in \boldsymbol{i}} e^{-\alpha y_i f_\theta(i)}},$$

*where $\alpha = \frac{1}{2}\ln(\frac{1-\epsilon}{\epsilon})$ and $\epsilon$ is the classification error of $f_\theta$. $p_{lpsc}(i)$ can be computed in $O(|\boldsymbol{i}|)$ time-complexity.*

Thereby, AdaBoost efficiently optimizes the LPSC problem. Notably, LPSC can be reduced to *different* boosting algorithms under different assumptions. In Section 5.4, we evaluate the performance of *AdaBoost* Freund & Schapire (1997), *LogitBoost* Friedman et al. (2000) and $L_2$-*Boost* Zheng & Liu (2012) for LPSC. We denote the LPSC-encoded privacy $p_{lpsc}(i, y)$ on aligned training data as $\mathcal{D}_{lpsc} = (\boldsymbol{w}, \boldsymbol{r})$, with sample weights $\boldsymbol{w}$ for $p_{lpsc}(i)$ and residuals $\boldsymbol{r}$ for $p_{lpsc}(y|i)$.

## 4.3 VFGBoost Framework

Based on our finding that gradient boosting solves the LPSC problem, we propose a novel Vertical Federated Gradient Boosting framework, VFGBoost, to efficiently and flexibly protect label privacy in VFL. VFGBoost leverages gradient boosting to solve the LPSC problem in an offline phase. However, as shown in Fig. 3, **LPSC only provides upper-bounded privacy protection without compromising utility**, which is upper-bounded by $p_{act}(i, y)$ and may not meet practical privacy requirements. Therefore, VFGBoost further incorporates *adversarial training* to enable additional and flexible privacy enhancement by sacrificing utility in the federated training phase. It should be noted that the integration of LPSC with other perturbation methods is also a valid approach for achieving additional privacy enhancement, as evaluated in Section 5.3.

To achieve offline LPSC (Fig. 2, phase 1), VFGBoost leverages gradient boosting to compute the re-weight residuals $\mathcal{D}_{lpsc}$. After LPSC, we shift the learning target from ground-truth labels to residuals with re-weighted samples. In the federated training phase (Fig. 2, phase 2), all parties collaboratively train a federated model $h_{fed}$ to fit the re-weighted residuals $\mathcal{D}_{lpsc}$ as follows:

$$h_{fed}(i) = g_\lambda \left( \{h_{\psi_k}(i)\}_{k=1}^K \right), \tag{5}$$

where $g_\lambda$ is the aggregation top model trained by the active party $P_0$, and $h_{\psi_k}(i)$ denotes $h_{\psi_k}(\boldsymbol{x}_{k,i})$ from $P_k$, for simplicity. The overall VFGBoost framework $f_{VFGBoost}$ can be expressed as:

$$f_{VFGBoost}(i) = f_\theta(i) + \alpha \cdot h_{fed}(i),$$

where $\alpha > 0$ represents the weight of the aggregated residuals.

### 4.3.1 Learning Objectives

The training procedure has two objectives: 1) utility objective $\mathcal{L}_{util}$ to fit the LPSC-encoded results $\mathcal{D}_{lpsc}$, and 2) privacy objective $\mathcal{L}_{priv}$ to further enhance label privacy via adversarial training.

**LPSC utility objective**. The utility objective trains the federated model $h_{fed}$ in Eq. 5 to fit LPSC-encoded re-weighted residuals $\mathcal{D}_{lpsc} = (\boldsymbol{w}, \boldsymbol{r})$ as follows:

$$\min_{\lambda, \{\psi_k\}_{k=1}^K} \sum_{i \in \boldsymbol{i}} w_i \cdot \mathcal{L}_{util} \left( r_i, h_{fed}(i) \right),$$

where $(w_i, r_i) \in \mathcal{D}_{lpsc}$ is the weight and residual of the $i$-th sample and $\mathcal{L}_{util}$ denotes utility loss.

**Adversarial privacy objective**. Given LPSC only provides upper-bounded privacy enhancement, we employ *adversarial training* to enable trading utility for additional privacy enhancement. Specifically, the active party mimics adversaries by conducting the PMC attack Fu et al. (2022). The active party $P_0$ trains adversarial models $\{a_{\phi_k}\}_{k=1}^K$ to attack each bottom model $\{h_{\psi_k}\}_{k=1}^K$, and in turn, trains the bottom models to defend against these attacks. Therefore, the adversarial training process can be formulated as a max-min optimization problem as follows:

$$\max_{\psi_k} \min_{\phi_k} \mathbb{E}_{i \sim p_{gt}(i)} \left[ \mathcal{L}_{priv(k)} \left( y_i, a_{\phi_k} \circ h_{\psi_k}(i) \right) \right] \quad s.t. \ \forall k \in [1, \dots, K],$$

where $\mathcal{L}_{priv(k)}$ denotes the privacy loss for passive party $P_k$.

In summary, the overall objective is to solve the following max-min optimization problem:

$$\min_{\lambda, \{\psi_k\}_{k=1}^K} \max_{\{\phi_k\}_{k=1}^K} \left\{ \underbrace{\sum_{i \in \boldsymbol{i}} w_i \cdot \mathcal{L}_{util} \left( r_i, h_{fed}(i) \right)}_{\text{LPSC utility objective}} - \beta \cdot \underbrace{\sum_{k=1}^K \sum_{i \in \boldsymbol{i}} \frac{1}{|\boldsymbol{i}|} \cdot \mathcal{L}_{priv(k)} \left( y_i, a_{\phi_k} \circ h_{\psi_k}(i) \right)}_{\text{Adversarial privacy objective}} \right\}, \tag{6}$$

where $\beta \geq 0$ is a small hyperparameter to control privacy-utility trade-off. **A non-zero $\beta$ enables the trade-off of utility for additional privacy enhancement, building on the inherent, yet upper-bounded, privacy provided by LPSC.** Algorithm 1 outlines the training process of VFGBoost.

---

**Algorithm 1** VFGBoost framework

---

**Require:** Local data $\mathcal{D}^{loc} = \{\boldsymbol{i}^{loc}, \boldsymbol{X}_0^{loc}, \boldsymbol{y}^{loc}\}$, and aligned data $\mathcal{D} = \{\boldsymbol{i}, \boldsymbol{X}_0, \ldots, \boldsymbol{X}_K, \boldsymbol{y}\}$.
    ▷ **Phase 1: Label privacy source coding (LPSC)**
  1:   Active party $P_0$ learns $p_{act}(i, y)$ by training $f_\theta$ on $\mathcal{D}^{loc}$ via Eq. 3.
  2:   Active party $P_0$ optimizes $p_{lpsc}(i, y)$ by computing weight-residual $\mathcal{D}_{lpsc} = (\boldsymbol{w}, \boldsymbol{r})$.
    ▷ **Phase 2: Federated training**
  3:   $P_0$ initializes $\lambda$ and $\{\phi_k\}_{k=1}^K$. Passive parties $\{P_k\}_{k=1}^K$ initialize $\{\psi_k\}_{k=1}^K$, respectively.
  4: **for** each batch of samples with IDs $\boldsymbol{b} \subset \boldsymbol{i}$ **do**
      ▷ **Loss Computation**
  5:     $\{P_k\}_{k=1}^K$ compute $\{\hat{\boldsymbol{r}}_k = h_{\psi_k}(\boldsymbol{b})\}_{k=1}^K$ and send to $P_0$.
  6:     $P_0$ computes $h_{fed}(\boldsymbol{b})$ via Eq. 5, then computes $\mathcal{L}_{util}$ and $\{\mathcal{L}_{priv(k)}\}_{k=1}^K$, via Eq. 6.
      ▷ **Model Update**
  7:     $P_0$ updates the aggregation top model $\lambda$ and adversarial models $\{\phi_k\}_{k=1}^K$ via gradients.
  8:     $\{P_k\}_{k=1}^K$ update bottom models $\{\psi_k\}_{k=1}^K$ via gradients.
  9: **end for**
**Ensure:** Local model $\theta$, top model $\lambda$, bottom models $\{\psi_k\}_{k=1}^K$.

---

## 5 EXPERIMENTS

### 5.1 EXPERIMENTAL SETTING

**Datasets.** We evaluate our proposed VFGBoost on four real-world datasets, including two widely used recommendation click-through rate (CTR) prediction datasets: Criteo[1] and Avazu[2], and two healthcare datasets: MIMIC-III Johnson et al. (2016) and Cardio. Each dataset is partitioned into five (Avazu) or seven (others) parties. We defer detailed descriptions of the datasets in Appendix F.1.

**Implementation.**[3] Without specification, we use *LogitBoost* Friedman et al. (2000) for LPSC. $\mathcal{D}_{lpsc}$ is computed following Table 6. We adopt DeepFM Guo et al. (2017) for both local and bottom models on Criteo and Avazu. We use MLP for both local and bottom models on MIMIC-III and Cardio datasets. We defer details of hyperparameter choices and platform descriptions in Appendix F.2.

**Compared Methods.** For fair comparisons, we select a set of label privacy protection methods applicable in VFL as baselines. Cryptographic approaches are not included due to their expensive communication and computational cost. 1) **FE-VFL** Sun et al. (2022) trains a top model to directly predict labels using forward embeddings, while simultaneously minimizing the distance correlation between the forward embeddings and the labels. 2) **CoAE** Zou et al. (2022) trains a deterministic mapping function that transforms original labels to surrogate labels. The bottom models are trained to predict the surrogate labels. 3) **MID** Zou et al. (2023) employs a VAE-based MI estimator Alemi et al. (2016) to explicitly estimate and minimize the entropy of the forward embedding during training. 4) **LabelDP** Ghazi et al. (2021) leverages a random response mechanism to randomly flip labels to generate perturbed gradients. 5) **Marvell** Li et al. (2022) uses adapted Gaussian noise to perturb the gradients, so that the distribution difference of positive and negative class's gradients are eliminated. 6) **LPSC+LabelDP** combines our gradient boosting-based offline LPSC with training-phase LabelDP Ghazi et al. (2021). 7) **LPSC+Marvell** integrates our gradient boosting-based offline LPSC with training-phase Marvell Li et al. (2022).

**Metrics.** We evaluate our method against baselines regarding utility and privacy. We use the AUC (Area Under ROC curve) metric in our experiments. 1) **Utility**: To gauge the utility of the federated models, we compute the AUC of the federated model (FL-AUC) on fully aligned test data. Higher values of FL-AUC indicate superior model utility. 2) **Privacy**: We evaluate the effectiveness of various defense approaches using three label privacy attacks: the Norm attack Li et al. (2022), Spectral attack Tran et al. (2018), and Passive Model Completion (PMC) attack Fu et al. (2022). For privacy evaluation, we calculate the average AUC of the label predictions made by the passive parties, which we refer to as label leakage AUC (LL-AUC). A low LL-AUC value, close to 0.5, signifies strong privacy protection.

---

[1]https://labs.criteo.com/category/dataset/

[2]https://www.kaggle.com/c/avazu-ctr-prediction

[3]The code is available at https://anonymous.4open.science/r/VFGBoost-D62D

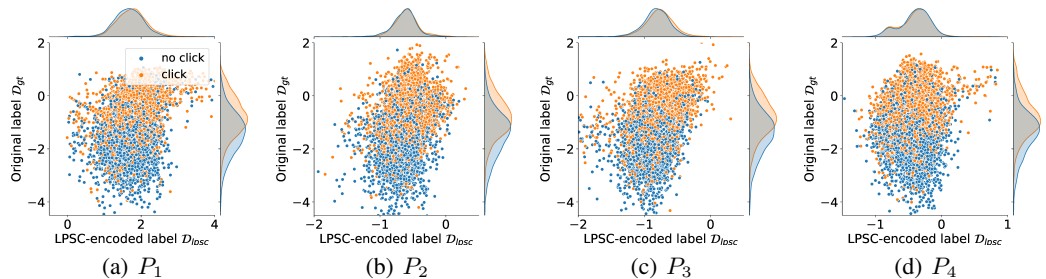

(a) $P_1$   (b) $P_2$   (c) $P_3$   (d) $P_4$

Figure 4: Distributions of passive parties' output logits by fitting original labels $p_{gt}$ v.s. LPSC-encoded labels $p_{lpsc}$ for the first four passive parties on the Criteo dataset. (Adversarial loss $\beta = 0$.)

## 5.2 LPSC Protects Label Privacy Without Compromising Utility

We first evaluate the protection quality of gradient boosting-based LPSC. Specifically, in the federated training phase, we train passive parties' bottom models to fit the LPSC-encoded labels $p_{lpsc}(i, y)$ and original ground-truth labels $p_{gt}(i, y)$, respectively.

Table 2 presents the LL-AUC against Norm, Spectral, and PMC attacks and the FL-AUC on four datasets. The results reveal that the LL-AUC of LPSC against three attacks is significantly lower than that of the original labels, indicating that LPSC provides strong label privacy protection.

Table 2: The comparison of privacy and utility of VFL fitting labels $p_{gt}$ v.s. LPSC $p_{lpsc}$ against Norm, Spectral, and PMC attacks. ↑ means desirable directions. $\beta = 0$.

| Dataset | Target | Privacy (LL-AUC) ↓ | | | Utility ↑ |
|---|---|---|---|---|---|
| | | Norm | Spectral | PMC | FL-AUC |
| Criteo | Label | 0.673 | 0.689 | 0.718 | 0.768 |
| | LPSC | **0.533** | **0.538** | **0.571** | 0.766 |
| Avazu | Label | 0.647 | 0.668 | 0.695 | 0.749 |
| | LPSC | **0.541** | **0.555** | **0.577** | 0.751 |
| MIMIC-III | Label | 0.636 | 0.653 | 0.681 | 0.768 |
| | LPSC | **0.528** | **0.535** | **0.558** | 0.768 |
| Cardio | Label | 0.582 | 0.618 | 0.640 | 0.721 |
| | LPSC | **0.533** | **0.542** | **0.567** | 0.720 |

Meanwhile, the FL-AUC of LPSC is comparable to that of the original labels, implying that LPSC barely sacrifices model utility. This confirms that LPSC can effectively protect label privacy without compromising utility. The PMC attack outperforms Norm and Spectral attacks in LL-AUC values, indicating its heightened threat. This stems from PMC's ability to utilize labeled samples for top model reconstruction. Thus, we employ PMC for subsequent label privacy evaluations.

Fig. 4 visualizes the output logits distributions of four passive parties by training with or without LPSC. The complete figures including all six passive parties can be found in the Appendix G.1. The top-side distributions in each sub-figure show that, with LPSC, the logits distributions of the two classes almost overlap and are hard to differentiate. In contrast, without LPSC, the right-side distributions in each sub-figure reveal significant differences in the distributions of the two classes, implying label privacy leakage. Our empirical findings are supported by the theoretical guarantee in Theorem 1, which justifies our observation that it is more challenging to distinguish the output distributions between classes when the bottom models are trained with LPSC-encoded labels.

## 5.3 Privacy-Utility Trade-off Comparison

Fig. 5 shows the privacy-utility trade-off curves on four datasets. The X-axis indicates the label leakage AUC (LL-AUC), and the Y-axis indicates the AUC of the federated model prediction (FL-AUC). An ideal trade-off should have a large FL-AUC and a small LL-AUC, thus residing in the upper-left corner of Fig. 5. Our VFGBoost is the closest to the ideal trade-off on all four datasets. We discuss how offline LPSC and training-phase adversarial training in VFGBoost improve the privacy-utility trade-off in the following, respectively.

**Label Privacy Source Coding.** To explore the effectiveness of LPSC, we compare LPSC-enhanced baselines (i.e., LPSC+LabelDP and LPSC+Marvell) with their counterparts without LPSC (i.e., LabelDP and Marvell). As shown in Fig. 5, LPSC significantly improves the privacy-utility trade-off of existing perturbation baselines by pushing the top-side of each curve leftwards on each dataset. Without any training-phase perturbation (the top-right end of each curve), LPSC leads to significant LL-AUC decline with negligible FL-AUC decline on each dataset, implying that it protects

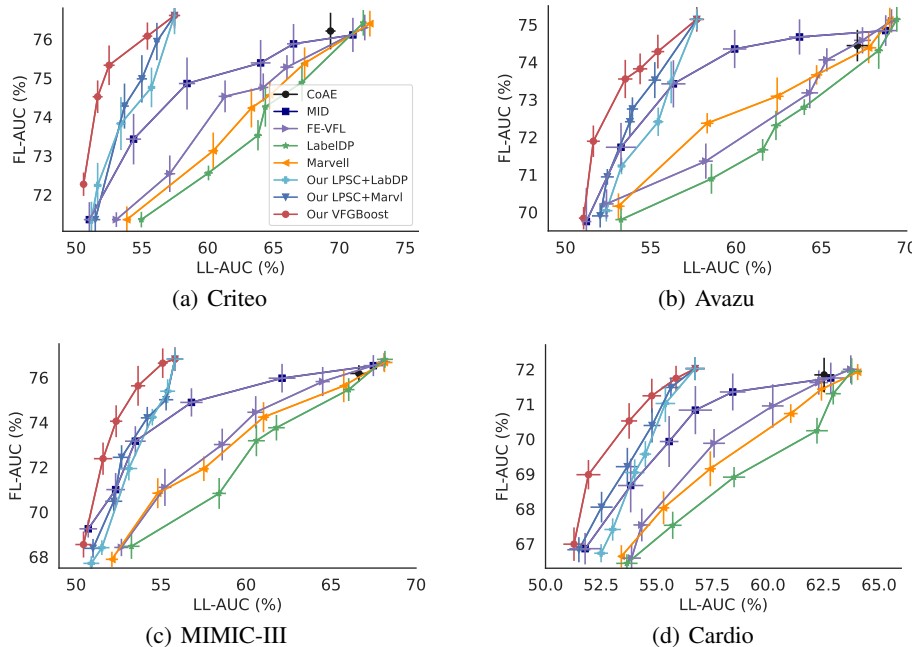

Figure 5: Privacy-utility trade-off of different label protection methods against the PMC attack on four datasets. All methods have the same dimension of forward embedding. Note that LPSC+LabDP and LPSC+Marvl are our LPSC combined with LabelDP and Marvell, respectively.

label privacy without sacrificing utility. This empirical observation is also justified by the theoretical guarantee in Theorem 1. Therefore, LPSC can be easily integrated with different training-phase perturbation methods for privacy-utility trade-off improvement.

**Adversarial Training.** To investigate the effectiveness of adversarial training, we compare VFG-Boost with two LPSC-enhanced baselines (i.e., LPSC+Marvell and LPSC+LabelDP). As shown in Fig. 5, we can observe that the trade-off curves of VFGBoost are closer to the upper-left corner than those of two LPSC-enhanced baselines on each dataset, indicating that VFGBoost outperforms them with big margins. This validates the effectiveness and superiority of adversarial training in VFGBoost for privacy-utility trade-off.

## 5.4 IMPACT OF GRADIENT BOOSTING ALGORITHMS ON LPSC

We compare the impact of different gradient boosting algorithms on LPSC, including AdaBoost, Logit-Boost, and $L_2$-Boost. For each boosting algorithm, $p_{lpsc}(i, y)$ is computed following Table 6 in Appendix C.1. AdaBoost updates the sample-weights $\boldsymbol{w}_i = p_{lpsc}(i)$ based on the classification error of the local model $f_\theta$. While, LogitBoost and $L_2$-Boost assign residuals $\boldsymbol{r}_i = p_{lpsc}(y|i)$ based on the negative gradient of the log-likelihood loss and the mean-

Table 3: The comparative AUC results of different gradient boosting algorithms. (No adversarial training i.e., $\beta = 0$.)

| Dataset | AUC | AdaBoost | LogitBoost | $L_2$-Boost |
|---------|-----|----------|------------|-------------|
| Criteo  | FL ↑ | 0.765 | **0.766** | 0.760 |
|         | LL ↓ | 0.584 | **0.571** | 0.603 |
| Avazu   | FL ↑ | **0.752** | 0.751 | 0.748 |
|         | LL ↓ | 0.582 | **0.577** | 0.592 |

square error loss, respectively. Table 3 shows the privacy-utility trade-off of different gradient boosting algorithms on Criteo and Avazu datasets. We find that LogitBoost is more effective for LPSC than the others in terms of the privacy-utility trade-off.

We defer additional experiments on **label privacy protection during training, feature privacy protection, and model-agnosticism** in Appendix G.

## 6 CONCLUSION

We focus on protecting label privacy in VFL without sacrificing utility and formulate the LPSC problem for offline-phase cleansing. Our analysis confirms that *gradient boosting* effectively tackles the LPSC problem, leading to the proposed VFGBoost framework. Incorporating *adversarial training*, VFGBoost further enables a nuanced privacy-utility trade-off. Experimental results on four datasets demonstrate the efficacy of LPSC and the superiority of our VFGBoost framework.

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

# A  NOTATIONS

In our discussions and formulations throughout this paper, we use several notations for clarity and brevity. A summary of these notations and their corresponding descriptions is provided in Table 4.

Table 4: Table of notations

| Notation | Description |
|---|---|
| $p_{gt}(i, y)$ | Ground-truth label privacy (ID-label joint distribution) |
| $p_{act}(i, y)$ | Active party learned label privacy |
| $p_{lpsc}(i, y)$ | LPSC-encoded label privacy |
| $\mathcal{D}_{lpsc} = (\boldsymbol{w}, \boldsymbol{r})$ | LPSC-encoded results with weights $\boldsymbol{w}$ and residuals $\boldsymbol{r}$ |
| $\boldsymbol{X}_k$ | Aligned feature matrix of party $P_k$ |
| $\boldsymbol{y}$ | Aligned vector of labels of active party $P_0$ |
| $\boldsymbol{i}$ | Aligned vector of sample IDs for aligned training data |
| $\boldsymbol{i}^{loc}, \boldsymbol{y}^{loc}, \boldsymbol{X}_0^{loc}$ | Active party $P_0$'s local IDs, labels and features |
| $f_\theta$ | Active party's local model |
| $h_{\text{fed}}$ | All passive parties' federated model |
| $h_{\psi_k}$ | Passive party $P_k$'s bottom model |
| $a_{\phi_k}$ | Adversarial top model for $P_k$ |
| $g_\lambda$ | Top model |
| $f_{VFGBoost}$ | Overall VFGBoost model |
| $\alpha$ | Weight of $h_{fed}$ predicted residual |
| $\beta$ | Weight of adversarial loss |
| $\mathcal{L}_{util}$ | LPSC utility loss |
| $\mathcal{L}_{priv(k)}$ | Adversarial privacy loss for $P_k$ |

# B  THREAT MODEL AND PRIVACY ANALYSIS

In this section, we provide a comprehensive analysis of privacy threats in Vertical Federated Learning (VFL), underscoring our commitment to safeguarding sensitive information. Our focus primarily lies on the threat model for label privacy protection, pivotal to the integrity of VFL systems. We begin by formally and rigorously defining the label privacy threat model (B.1). Subsequently, we delve into the intricate relationship between attack error and mutual information privacy (MIP)(B.2), illustrating how these concepts form the backbone of our privacy strategy. The subsection "Mutual Information Privacy Guarantee" (B.3) offers a rigorous validation of our approach, providing theoretical assurances of privacy preservation. Finally, we draw a parallel between MIP and Differential Privacy (DP) in "Comparison of MIP v.s. DP" (B.4,) highlighting the distinct advantages and considerations of our chosen methodology in the context of VFL. This comprehensive approach not only addresses label privacy from multiple angles but also situates our work within the broader landscape of privacy research.

## B.1  LABEL PRIVACY THREAT MODEL

**Adversary's capabilities**. A semi-honest non-colluding passive party $P_k$ has a bottom model $h_{\psi_k}(\cdot)$ which outputs forward embeddings $h_{\psi_k}(i) = h_{\psi_k}(\boldsymbol{x}_{k,i})$ of the $i$-th sample in its features $\boldsymbol{X}_k$. However, $P_k$ has *no* prior knowledge of the active party's data $\{\boldsymbol{i}, \boldsymbol{y}, \boldsymbol{X}_0\}$.

**Adversary's objective**. The adversary's objective is to minimize the *expected error* of label estimation on a dataset by optimizing an attack function $A(\cdot) \in \mathbb{A}$ as follows:

$$\min_{A \in \mathbb{A}} \mathbb{R}_{p_{gt}(i,y)}(A \circ h_{\psi_k}) = \min_{A \in \mathbb{A}} \mathbb{E}_{i \sim p_{gt}(i)}[D_{\text{KL}}(p_{gt}(y|i)||A(h_{\psi_k}(i)))].$$

We use norm attack Li et al. (2022), spectral attack Tran et al. (2018), or passive model completion (PMC) attack Fu et al. (2022) to build the attack function $A(\cdot)$ in our paper.

Table 5 summarizes the threat model of label privacy protection in VFL.

| Adversary | Passive party $P_k$ |
|---|---|
| Threat model | Semi-honest, non-colluding |
| Adversary's knowledge | Forward embeddings $h_{\psi_k}(i)$ from bottom model $h_{\psi_k}$. No prior knowledge of $p_{gt}(i, y, X_0)$, i.e., $\{\boldsymbol{i}, \boldsymbol{y}, \boldsymbol{X}_0\}$. |
| Adversary's objective | Minimize the error $\mathbb{R}_{p_{gt}(i,y)}(A \circ h_{\psi_k})$ in Equation 1 by optimizing attack function $A(\cdot)$. |

Table 5: Label privacy threat model

## B.2 Bridging Attack Error and Mutual Information Privacy

As outlined in Section 3, our threat model aligns with the concept of mutual information privacy (MIP). This subsection delves into *how the threat model naturally leads to the adoption of MIP as our privacy definition*.

The adversary's goal, as defined in Equation 1, is to minimize the expected estimation error regarding the ground-truth joint distribution $p_{gt}(i, y)$. Given the adversary has *no prior knowledge* about $p_{gt}(i, y)$ and access only to forward embeddings, the private label information to protect **is and only is** the *ID-label joint distribution $p_{gt}(i, y)$*.

To understand how our approach fits within the broader landscape of privacy definitions and mechanisms, we first define what constitutes a privacy definition and a privacy mechanism in the context of our work:

**Definition 1** (Privacy Definition Kifer & Lin (2012)). *Given an input space $\mathbb{I}$, a privacy definition is a set of randomized algorithms with common input space $\mathbb{I}$. We say that these randomized algorithms satisfy the privacy definition.*

**Definition 2** (Privacy Mechanism Kifer & Lin (2012)). *A privacy mechanism $\mathcal{M}$ is a randomized algorithm $\mathcal{M}$ that satisfies a privacy definition.*

In our context, the privacy mechanism $\mathcal{M}$ inputs active party's dataset distribution $p_{gt}(i, y, X_0)$, as defined by $\mathcal{D}^{loc} = \{\boldsymbol{i}^{loc}, \boldsymbol{y}^{loc}, \boldsymbol{X}_0^{loc}\}$ and generates a new joint distribution:

$$p_{\mathcal{M}}(i, y) = \mathcal{M}(p_{gt}(i, y, X_0)).$$

During federated training, the bottom model $h_{\psi_k}$ is trained to align with $p_{\mathcal{M}}(i, y)$. Consequently, the optimal attack error can be expressed as:

$$\mathbb{R}_{p_{gt}(i,y)}(p_{\mathcal{M}}(i, y)) = \min_{A \in \mathbb{A}} \mathbb{E}_{i \sim p_{gt}(i)}[D_{\mathrm{KL}}(p_{gt}(y|i)||A(h_{\psi_k^*}(i)))],$$
$$\text{where } \psi_k^* = \arg\min_{\psi_k} \mathbb{E}_{i \sim p_{\mathcal{M}}(i)}[D_{\mathrm{KL}}(p_{\mathcal{M}}(y|i)||g(h_{\psi_k}(i)))].$$

To enhance label privacy, $\mathcal{M}$ should be designed to maximize this attack error $\mathbb{R}_{p_{gt}(i,y)}(p_{\mathcal{M}}(i, y))$. We observe that maximizing $\mathbb{R}_{p_{gt}(i,y)}(p_{\mathcal{M}}(i, y))$ is equivalent to minimizing the mutual information between $p_{gt}(i, y)$ and $p_{\mathcal{M}}(i, y)$, i.e.,

$$\max_{p_{\mathcal{M}}(i,y)} \mathbb{R}_{p_{gt}(i,y)}(p_{\mathcal{M}}(i, y)) \iff \min_{p_{\mathcal{M}}(i,y)} I(p_{gt}(i, y); p_{\mathcal{M}}(i, y)),$$

where $I(\cdot; \cdot)$ denotes the mutual information. Thus, a privacy mechanism $\mathcal{M}$ minimizing mutual information $I(p_{gt}(i, y); p_{\mathcal{M}}(i, y))$ equivalently maximizes the attack error, protecting label privacy. Accordingly, our approach aligns with the principle of $\epsilon$-*mutual information privacy ($\epsilon$-MIP)*:

**Definition 3** ($\epsilon$-Mutual Information Privacy). *According to Wang et al. (2016), a mechanism $\mathcal{M}$ satisfies $\epsilon$-MIP for some $\epsilon \in \mathbb{R}^+$ if, for any input $X$, the mutual information between $X$ and the output $Y = \mathcal{M}(X)$ is bounded by $\epsilon$ bits, formally:*

$$I(X; Y) \leq \epsilon \text{ bits}.$$

## B.3 Mutual Information Privacy Guarantee

We prove that our proposed LPSC satisfies the $\epsilon$-MIP.

**Theorem 1** (Privacy Guarantee.) *LPSC satisfies $\epsilon$-mutual information privacy ($\epsilon$-MIP). The privacy leakage is bounded by $\epsilon = H(p_{gt}(i, y)|p_{act}(i, y))$, the conditional entropy of the ground-truth label distribution $p_{gt}(i, y)$ given the active party's label distribution $p_{act}(i, y)$. Formally,*

$$I(p_{gt}(i, y); p^*_{lpsc}(i, y)) \leq \epsilon \text{ bits,}$$

*where $p^*_{lpsc}(i, y)$ represents the optimal solution of the LPSC problem.*

*Proof.* We approach the LPSC problem as defined in Equation 2, optimizing $p_{lpsc}(i, y)$ with respect to:

$$p^*_{lpsc}(i, y) = \underset{p_{lpsc}(i,y)}{\arg\max} \, I(p_{gt}(i, y); p_{lpsc}(i, y))$$

$$s.t. \ \ I(p_{act}(i, y); p_{lpsc}(i, y)) = 0.$$

The key constraint is that $p_{act}(i, y)$ and $p_{lpsc}(i, y)$ must remain independent, which implies that mutual information between $p_{gt}(i, y)$ and $p_{lpsc}(i, y)$ excludes any shared information with $p_{act}(i, y)$. Analytically, we express this as:

$$I(p_{gt}(i, y); p^*_{lpsc}(i, y))$$
$$= I(p_{gt}(i, y); p^*_{lpsc}(i, y)|p_{act}(i, y))$$
$$\leq H(p_{gt}(i, y)|p_{act}(i, y))$$
$$= \epsilon \text{ bits,}$$

where $H(p_{gt}(i, y)|p_{act}(i, y))$ represents the conditional entropy, or the remaining uncertainty in $p_{gt}(i, y)$ after observing $p_{act}(i, y)$.

Therefore, the solution $p^*_{lpsc}(i, y)$ satisfies the $\epsilon$-MIP criterion, effectively bounding the mutual information and safeguarding label privacy in accordance with the $\epsilon$-MIP definition. $\square$

### B.4 COMPARISON OF MIP V.S. DP

Differential privacy (DP) Dwork et al. (2006), a well-established privacy definition in privacy-preserving machine learning, offers robust privacy guarantees by ensuring that the output of a mechanism does not significantly change with the alteration of a single record in the dataset. In contrast, mutual information privacy (MIP), which we adopt in this paper, focuses on limiting the mutual information between the input and output of a privacy mechanism.

A related concept, $\epsilon$-mutual-information differential privacy (MI-DP) Cuff & Yu (2016), bridges these two definitions. It is defined as follows:

**Definition 4** ($\epsilon$-Mutual Information Differential Privacy Cuff & Yu (2016)). *A mechanism $\mathcal{M}$ satisfies $\epsilon$-mutual-information differential privacy for some $\epsilon \in \mathbb{R}^+$ if for any neighboring inputs $X, X'$, the conditional mutual information between $X$ and $Y = \mathcal{M}(X)$ conditioned on $X'$ satisfies*

$$I(X; Y|X') \leq \epsilon \text{ bits.}$$

$\epsilon$-MI-DP is shown to be weaker than $\epsilon$-DP but stronger than $(\epsilon, \delta)$-DP, offering a middle ground in terms of privacy strength Cuff & Yu (2016).

The fundamental difference between MIP and DP, including its variant MI-DP, lies in their underlying *threat models*. DP operates under a strong adversary assumption, considering adversaries that have access to neighboring input databases. In contrast, MIP is designed under the assumption that the adversary lacks prior knowledge of the database, which aligns more closely with the threat model in our VFL setting.

Given the assumption in our VFL setting where an adversary has *no* prior knowledge of the active party's data $\{i, y, X_0\}$, MIP emerges as a more fitting choice. Our threat model, as elaborated in Section 3, coincides with the principles of MIP, making it a natural fit for our research.

Wang et al. (2016) reveals a fundamental connection privacy between mutual information and differential privacy bridged by the notion of *identifiability*. In future work, exploring the trade-offs and potential synergies between these privacy models could further enhance the applicability of privacy-preserving techniques in diverse machine learning scenarios.

## C BACKGROUND

### C.1 GRADIENT BOOSTING

Gradient boosting Freund & Schapire (1997) is a classic algorithm in ensemble learning and is known for reducing the *bias* of a weak learner. In VFL, the generalization error of the active party's local model primarily comes from *bias* instead of variance. This is because the active party's model complexity is restricted by insufficient features. Therefore, we can use boosting to reduce the bias of the local model by training passive parties to fit weighted-residuals. The predicted residuals are then added to active party's local model to reduce the bias.

Boosting is a functional gradient descent method for function estimation Friedman (2001); Friedman et al. (2000), which is a "stagewise, additive model." Consider the problem of function estimation

$$f^*(\boldsymbol{x}) = \arg\min_f \mathbb{E}_{(\boldsymbol{x},y)\sim\mathcal{D}} \left[ \mathcal{L}\left(y, f(\boldsymbol{x})\right) \right], \tag{7}$$

where $\mathcal{L}(\cdot,\cdot)$ is a differentiable and convex loss function. The generic gradient boosting algorithm is shown in Algorithm 2. Notably, the base learners can be *any gradient-based model* such as DNN Guo et al. (2017) and decision trees Freund & Schapire (1997).

---

**Algorithm 2** Gradient Boosting Freund & Schapire (1997)

---

**Require:** Data $\{(\boldsymbol{x}_i, y_i)\}_{i=1}^n$, loss function $\mathcal{L}(\cdot,\cdot)$
1: Train $f^{(0)} = \arg\min_{f^{(0)}} \frac{1}{n} \sum_{i=1}^n \mathcal{L}(y_i, f^{(0)}(\boldsymbol{x}_i))$.
2: **for** iteration $m \in [1, \ldots, M]$ **do**
3:      Update residuals $\mathbf{r}_i = -\frac{\partial \mathcal{L}(y_i, \hat{y}_i)}{\partial \hat{y}_i}|_{\hat{y}_i = f^{(m-1)}(\boldsymbol{x}_i)}$ of training data.
4:      Train $h^{(m)} = \arg\min_{h^{(m)}} \frac{1}{n} \sum_{i=1}^n \mathcal{L}(\mathbf{r}_i, h^{(m)}(\boldsymbol{x}_i))$
5:      Update the model by $f^{(m)}(\cdot) = f^{(m-1)}(\cdot) + \alpha h^{(m)}(\cdot)$
6: **end for**
7: **return** $f^{(M)}(\cdot)$

---

As shown in Table 6, gradient boosting algorithms generate weight-residual distributions of the dataset, given different loss functions. In Section 4.2, we will demonstrate that gradient boosting solves our proposed LPSC problem and enhances label privacy without compromising utility.

Table 6: Gradient boosting generates weight-residual joint distributions for samples. Weight is not normalized.

| Methods | Loss | Weight $p(i)$ | Residual $p(y\|i)$ |
|---|---|---|---|
| AdaBoost Freund & Schapire (1997) | $\exp(-y \cdot \hat{y})$ | $\exp(-y \cdot \hat{y})$ | $y$ |
| LogitBoost Friedman et al. (2000) | $\ln(1 + \exp(-y \cdot \hat{y}))$ | $\hat{y}(1 - \hat{y})$ | $\frac{y-\hat{y}}{\hat{y}(1-\hat{y})}$ |
| $L_2$-Boost Bühlmann & Yu (2003) | $(y - \hat{y})^2/2$ | $1$ | $y - \hat{y}$ |

## D THEORETICAL ANALYSIS

### D.1 PROOF OF THEOREM 2

**Theorem 2**. *Assuming fixed conditional distribution $p_{lpsc}(y|i) = p_{gt}(y|i)$ and let $U$ denote a uniform distribution, the LPSC problem 1 can be reduced to:*

$$\min_{p_{lpsc}(i)} \quad D_{\mathrm{KL}}(p_{lpsc}(i) \,||\, U) \quad s.t. \sum_{i\in\boldsymbol{i}} p_{lpsc}(i) y_i f_\theta(i) = 0,$$

*where $i \in \boldsymbol{i}$ is the sample index of aligned training data with IDs $\boldsymbol{i}$.*

*Proof.* Given the assumption that the conditional distribution $p_{lpsc}(y|i) = p_{gt}(y|i)$, we first consider Eq. 2.

$$\max_{p_{lpsc}(i,y)} I(p_{gt}(i,y); p_{lpsc}(i,y))$$

$$\Longleftrightarrow \max_{p_{lpsc}(i,y)} I(p_{gt}(i)p_{gt}(y|i); p_{lpsc}(i)p_{lpsc}(y|i))$$

$$\Longleftrightarrow \max_{p_{lpsc}(i)} I(p_{gt}(i); p_{lpsc}(i))$$

$$\Longleftrightarrow \max_{p_{lpsc}(i)} I(p_{lpsc}(i); U)$$

$$\Longleftrightarrow \min_{p_{lpsc}(i)} D_{\mathrm{KL}}(p_{lpsc}(i)||U). \tag{8}$$

Eq. 8 minimizes a KL-divergence between $p_{lpsc}(i)$ and a uniform distribution $U$.

Recall the constraint of zero mutual information in Problem 1:

$$I(p_{act}(i,y); p_{lpsc}(i,y)) = 0. \tag{9}$$

Eq. 9 implies that $p_{act}(i,y)$ and $p_{lpsc}(i,y)$ are independent. To construct an independent $p_{lpsc}(i,y)$, we constrain the *correlation* of the two joint distributions $p_{act}(i,y)$ and $p_{lpsc}(i,y)$ as follows:

$$I(p_{act}(i,y); p_{lpsc}(i,y)) = 0$$

$$\Longrightarrow \sum_{i \in I} p_{act}(i,y)p_{lpsc}(i,y) = 0$$

$$\Longleftrightarrow \sum_{i \in I} p_{act}(i)p(y = f_\theta(i)|i) \cdot p_{lpsc}(i)p(y = y_i|i) = 0$$

$$\Longleftrightarrow \sum_{i \in I} \frac{1}{|I|}p(y = f_\theta(i)|i) \cdot p_{lpsc}(i) \cdot y_i = 0$$

$$\Longleftrightarrow \sum_{i \in I} p_{lpsc}(i)y_i f_\theta(i) = 0.$$

Therefore, the LPSC problem 1 can be reduced to:

$$\min_{p_{lpsc}(i)} D_{\mathrm{KL}}(p_{lpsc}(i) \| U) \quad s.t. \sum_{i \in I} p_{lpsc}(i)y_i f_\theta(i) = 0.$$

$\square$

## D.2 Proof of Theorem 3

**Theorem 3**. Schapire & Freund (2013) *The solution of the convex optimization problem Eq. 4 is equivalent to AdaBoost Freund & Schapire (1997):*

$$p_{lpsc}(i) = \frac{e^{-\alpha y_i f_\theta(i)}}{\sum_{i \in \boldsymbol{i}} e^{-\alpha y_i f_\theta(i)}},$$

*where $\alpha = \frac{1}{2}\ln(\frac{1-\epsilon}{\epsilon})$ and $\epsilon$ is the classification error of $f_\theta$. $p_{lpsc}(i)$ can be computed in $O(|\boldsymbol{i}|)$ time-complexity.*

*Proof.* Recall Eq. 4 in Theorem 2:

$$\min_{p_{lpsc}(i)} D_{\mathrm{KL}}(p_{lpsc}(i) \| U) \quad s.t. \sum_{i \in \boldsymbol{i}} p_{lpsc}(i)y_i f_\theta(i) = 0,$$

According to Eq. 4, we can compute this minimization by forming the Lagrangian:

$$\mathcal{L} = \sum_{i \in I} p_{lpsc}(i) \ln \frac{p_{lpsc}(i)}{p_{act}(i)} + \alpha \sum_{i \in I} p_{lpsc}(i)y_i f_\theta(i) + \mu(\sum_{i \in I} p_{lpsc}(i) - 1). \tag{10}$$

Here, $\alpha$ and $\mu$ are the Lagrange multipliers, and we have explicitly taken into account the constraint that

$$\sum_{i \in I} p_{lpsc}(i) = 1. \tag{11}$$

By computing derivatives and equating them with zero, we get that

$$0 = \frac{\partial \mathcal{L}}{\partial p_{lpsc}(i)} = \ln \left( \frac{p_{lpsc}(i)}{p_{act}(i)} \right) + 1 + \alpha y_i f_\theta(i) + \mu.$$

Given $p_{act}(i) = p_{gt}(i) \sim U$ is a uniform distribution, we have

$$p_{lpsc}(i) = \frac{1}{|I|} e^{-\alpha y_i f_\theta(i) - 1 - \mu}.$$

Note that $\mu$, an arbitrary constant, will be chosen to enforce Eq. 11, giving

$$p_{lpsc}(i) = \frac{e^{-\alpha y_i f_\theta(i)}}{Z}$$

where

$$Z = \sum_{i \in I} e^{-\alpha y_i f_\theta(i)}$$

is a normalization factor. Plugging into Eq. 10 and simplifying gives

$$\mathcal{L} = -\ln Z.$$

We optimize $\alpha$ to maximize $\mathcal{L}$ or, equivalently, to minimize $Z$. Thus, we get:

$$\alpha = \frac{1}{2} \ln(\frac{1 - \epsilon}{\epsilon}),$$

where $\epsilon$ is the weighted error rate of the weak learner $f_\theta(i)$. $\square$

## E    DETAILED EXPLANATIONS OF VFGBOOST ALGORITHM

Algorithm 1 outlines the full two-phase training process of VFGBoost. The first offline LPSC phase involves using a gradient boosting algorithm to compute $p_{lpsc}(i, y)$ that forms the learning target for federated training. In the subsequent federated training phase, passive parties $P_k$ compute forward embeddings $\hat{\boldsymbol{r}}_{\boldsymbol{k}} = h_{\psi_k}(\boldsymbol{b})$ for each batch, and them send to the active party. The active party $P_0$ then computes the utility loss $\mathcal{L}_{util}$ and privacy losses $\{\mathcal{L}_{priv(k)}\}_{k=1}^{K}$, and sends the residual gradients $\{\nabla \hat{\boldsymbol{r}}_k\}_{k=1}^{K}$ to the passive parties for model updates, and concurrently updates the aggregation top model $a_\lambda$ and the adversarial models $\{g_{\phi_k}\}_{k=1}^{K}$. Specifically, the gradients of the forward embeddings (residuals) $\hat{\boldsymbol{r}}_k = h_{\psi_k}(\boldsymbol{b})$ are computed as follows:

$$\nabla \hat{\boldsymbol{r}}_k = \frac{\partial \mathcal{L}_{util}}{\partial \hat{\boldsymbol{r}}_k} - \beta \cdot \frac{\partial \mathcal{L}_{priv(k)}}{\partial \hat{\boldsymbol{r}}_k},$$

where $\beta \geq 0$ is a small hyperparameter to control privacy-utility trade-off. **None-zero $\beta$ allows trading utility for further privacy enhancement, on the basis of the upper-bounded privacy given by LPSC.**

## F    DATA SET UP AND EXPERIMENTAL DETAILS

### F.1    DATASETS

We work with four datasets in our experiments: two recommendation datasets (Criteo and Avazu) and two healthcare datasets (MIMIC-III and Cardio). Table 7 shows the statistics of each dataset.

**Criteo**[4]. The Criteo dataset consists of ad click data over a week. For the Criteo dataset, each record contains 26 categorical input features and 13 real-valued input features. To prepare the data for our experiments, we first replace all missing categorical feature values with a single new category represented by an empty string, and replace all missing real-valued feature values with 0. We then convert each categorical feature value to a unique integer between 0 and the total number of unique categories, and linearly normalize each real-valued feature into the range [0, 1]. We randomly sample 10,000,000 records from the publicly provided Criteo training set and split the data into an 80%-20% train-test split for faster training and to generate privacy-utility trade-off comparisons. We randomly and evenly partition the features into 7 parts, for one active party and 6 passive parties.

**Avazu**[5]. Avazu contains 10 days of click logs. It has a total of 23 fields with categorical features including app ID, app category, device ID, etc. The missing categorical features are processed in the same way as the Criteo dataset. We use all available records in the dataset and randomly split the data into an 80%-20% train-test split. We randomly and evenly partition the categorical fields into 5 parties.

**MIMIC-III** Johnson et al. (2016) is a dataset designed for the in-hospital mortality prediction task, which involves predicting in-hospital mortality based on the first 48 hours of a patient's ICU stay. The dataset comprises 714 features and 20,000 records. To simulate multiple hospitals with shared services (features), we evenly split the features among seven parties at random. We use all available records in the dataset and randomly split the data into an 80%-20% train-test split. We randomly and evenly partition the features into 7 parts.

**Cardio**. The Cardio dataset comprises 246 real-valued features such as age, gender, diabetes, blood pressure, obesity, and more. These features were collected from 3,569 patients to predict whether a patient has cardiovascular disease. To simulate multiple hospitals with shared services (features), we evenly split the features among seven parties at random. We use all available records in the dataset and randomly split the data into an 80%-20% train-test split. We randomly and evenly partition the features into 7 parts.

| Dataset | # Train /Test | #Num. Feature | #Cate. Fields | #Num. Parties |
|---------|---------------|---------------|---------------|---------------|
| Criteo | 8,000,000/ 2,000,000 | 13 | 26 | 7 |
| Avazu | 8,000,000/ 2,000,000 | 0 | 21 | 5 |
| MIMIC-III | 16,912/ 4228 | 714 | 0 | 7 |
| Cardio | 2,856/ 713 | 246 | 0 | 7 |

Table 7: Statistics for all four datasets.

### F.2 MODEL ARCHITECTURE AND TRAINING DETAILS

**Model architecture details. [Criteo, Avazu]** We use a popular deep learning model DeepFM Guo et al. (2017) by default for online advertising. DeepFM is a hybrid model that combines factorization machines and deep neural networks for recommendation tasks. It has two main components: a factorization machine that captures pairwise feature interactions and a deep neural network that learns higher-order interactions and non-linear dependencies. The model takes input features and passes them through both components before concatenating the outputs and passing them to a final output layer. We follow the default model architecture configurations in the DeepCTR framework Shen (2017). All active and passive parties have the same deep model architecture. The embedding dimension is set as 4. The architectures of EDCN Chen et al. (2021), NFM He & Chua (2017) and WDL Cheng et al. (2016) used in the experiments are also the default model architecture configurations in the DeepCTR framework Shen (2017).

**[MIMIC-III, Cardio]** We use a 3-layer MLP model in each party to learn the mortality rate in MIMIC-III and the cardiovascular disease in Cardio. The dimension of each layer are [128, 64, 1].

**Model training details.** The models are optimized by Adam Kingma & Ba (2014). Table 8 shows the model training details on four datasets. We use the Adam optimizer Kingma & Ba (2014). The boosting model weight $\alpha$ to 1, the privacy coefficient $\beta$ to 0.05. We use 5-fold validation to

---

[4]https://labs.criteo.com/category/dataset/
[5]https://www.kaggle.com/c/avazu-ctr-prediction

determine early stopping. There are 50% samples aligned across all parties. All experiments were conducted on a system equipped with an Nvidia GTX 3080 GPU and 32 GB of RAM, running Ubuntu 22.04 as the operating system.

| Dataset | Optimizer | Lr | Batch Size | Epoch |
|---------|-----------|-----|-----------|-------|
| Criteo | Adam | $1e-4$ | 2,048 | 5 |
| Avazu | Adam | $5e-4$ | 4,096 | 5 |
| MIMIC-III | Adam | $5e-4$ | 16,912 | 200 |
| Cardio | Adam | $5e-4$ | 2,856 | 200 |

Table 8: Experimental settings for different datasets.

## G ADDITIONAL EXPERIMENTS

### G.1 LPSC PROTECTS LABEL PRIVACY

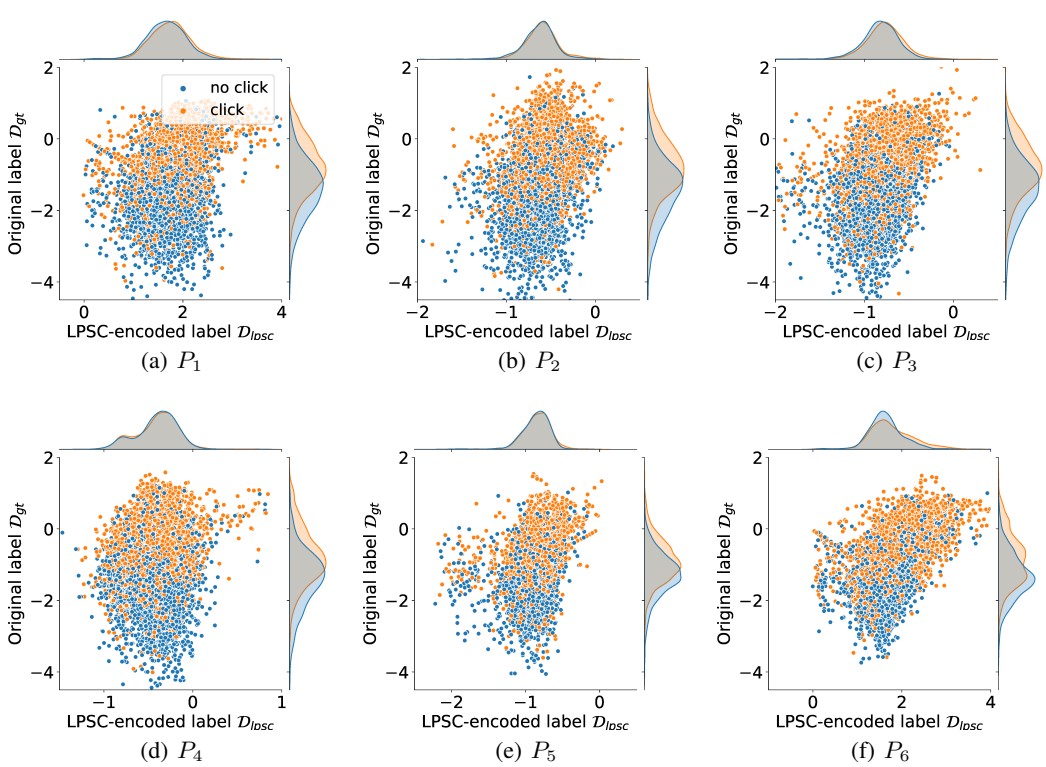

Figure 6: Distributions of passive parties' output logits by fitting original labels $p_{gt}(i, y)$ v.s. LPSC-encoded labels $p_{lpsc}(i, y)$ for all six passive parties on the Criteo dataset. (No adversarial training, i.e., $\beta = 0$.)

Fig. 6 demonstrates the distributions of all six passive parties' output logits by fitting original labels $p_{gt}(i, y)$ and LPSC-encoded labels $p_{lpsc}(i, y)$, respectively, on the Criteo dataset.

## G.2 Label Privacy Protection During Training

We study the label privacy protection from the forward embeddings in VFGBoost during the federated training phase. Fig. 7 demonstrates the distribution of LL-AUC values of all six passive parties' bottom models during training on the Criteo dataset. We observe that the LL-AUC values grow initially but gradually decrease towards $50\%$ as the round number increases. After training, the average AUC is $52.4\%$, which is close to the random guessing AUC $50\%$. Therefore, the forward embeddings in VFGBoost can protect label privacy during federated training.

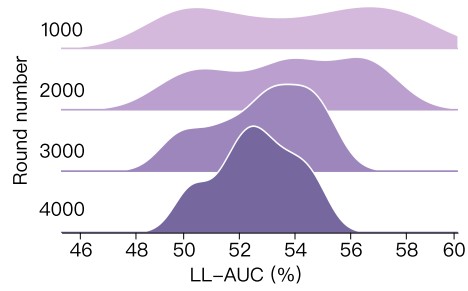

Figure 7: Distributions of the LL-AUC of all passive parties' bottom models during training. (From top to down.)

## G.3 Feature Privacy Protection

Interestingly, LPSC leads to natural feature privacy enhancement, which is not the primary goal of LPSC but a desirable side effect. The reason is that, by shifting the learning target from original labels $p_{gt}(i, y)$ to LPSC-encoded labels $p_{lpsc}(i, y)$, the active party no longer requires high-dimensional forward embeddings to maintain a complex deterministic top model to protect labels. Consequently, passive parties can send low-dimensional forward embeddings instead. This change eliminates the feature information leaked to the active party and enhances feature privacy protection.

Fig. 8 shows that lower-dimensional forward embeddings as in VFGBoost can provide stronger feature privacy protection against model inversion attack He et al. (2019) on the Avazu dataset, by following the settings of prior works Ye et al. (2022). The adversarial active party conducts the model inversion attack by training an MLP model to steal a private binary feature, "banner_pos." As shown in Fig. 8, we observe that when the embedding dimension is low (e.g., 1), the feature privacy leakage AUC is only $53.4\%$. However, when the embedding dimension is 32, the private feature can be accurately inferred with AUC=$92.8\%$. Therefore, high-dimensional embeddings may cause serious feature privacy leakage. However, due to offline LPSC, VFGBoost can protect feature privacy by exposing low-dimensional residuals.

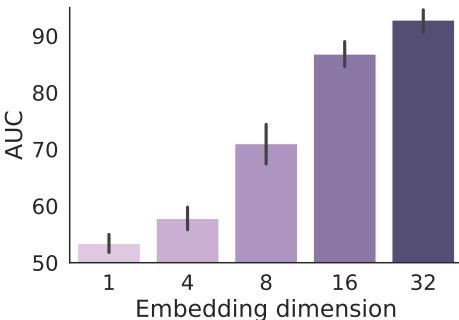

Figure 8: Feature privacy leakage against model inversion attack on the Avazu dataset. Our VFGBoost embedding dimension = 1.

## G.4 Impact of Heterogeneous Model Architectures

We study the impact of heterogeneous model architectures among parties in VFGBoost by replacing the passive parties' bottom models with EDCN Chen et al. (2021), NFM He & Chua (2017) and WDL Cheng et al. (2016), respectively. The active party's local model is fixed as a DeepFM. As shown in Table 9, with different model architectures, our proposed VFGBoost can effectively learn to fit $p_{lpsc}(i, y)$ and consistently achieves high FL-AUC and low LL-AUC. Therefore, VFGBoost is model-agnostic and applies to any gradient-based model.

Table 9: The comparative result of different model architectures of bottom models in VFGBoost.

| Dataset | AUC | EDCN | NFM | WDL |
|---------|-----|------|-----|-----|
| Criteo | FL-AUC (%) | 76.7 | 76.7 | 76.5 |
|        | LL-AUC (%) | 52.8 | 52.6 | 52.0 |
| Avazu | FL-AUC (%) | 75.2 | 75.1 | 74.9 |
|       | LL-AUC (%) | 53.6 | 53.4 | 53.3 |

### G.5  IMPACT OF LOCAL FEATURE QUALITY

To address concerns about our method's dependency on local feature quality, we conducted additional experiments on the Criteo dataset, examining the variability of the active party's feature set. We maintain a consistent partitioning of the dataset's 39 features across seven parties—one active and six passive. The active party's feature set is varied across experiments, with the number of features ranging from 1 to 20. We assess the impact of these variations on the model's performance by analyzing the active party's Local model AUC (Local-AUC), the Federated model AUC (FL-AUC), and the Label Leakage AUC (LL-AUC). Note that when the active party has no local features, the Local-AUC is 0.5, representing random guessing.

Table 10 illustrates that a minimal feature set in the active party correlates with lower Local-AUC and higher LL-AUC. As the number of features increases, Local-AUC improves, reducing LL-AUC and indicating enhanced privacy protection. The FL-AUC remains stable, affirming our method's robustness. This empirical observation validates the privacy guarantee in Theorem 1. This demonstrates our method's efficacy in balancing utility and privacy with different local feature settings.

Table 10: Impact of local features on LPSC performance in the Criteo dataset.

| # Local Features | Loc-AUC ($\pm$std) | FL-AUC ($\pm$std) | LL-AUC ($\pm$std) |
|------------------|--------------------|-------------------|-------------------|
| 0  | 0.5            | 76.79 ($\pm$0.34) | 72.07 ($\pm$0.33) |
| 1  | 60.72 ($\pm$0.39) | 76.86 ($\pm$0.28) | 66.24 ($\pm$0.34) |
| 2  | 64.68 ($\pm$0.47) | 76.48 ($\pm$0.32) | 63.34 ($\pm$0.47) |
| 4  | 70.91 ($\pm$0.48) | 76.79 ($\pm$0.38) | 61.67 ($\pm$0.48) |
| 5  | 71.31 ($\pm$0.34) | 76.62 ($\pm$0.24) | 58.73 ($\pm$0.42) |
| 8  | 72.26 ($\pm$0.38) | 76.39 ($\pm$0.27) | 56.16 ($\pm$0.48) |
| 10 | 74.37 ($\pm$0.35) | 76.51 ($\pm$0.34) | 54.83 ($\pm$0.51) |
| 15 | 74.13 ($\pm$0.52) | 76.38 ($\pm$0.43) | 54.77 ($\pm$0.38) |
| 20 | 75.04 ($\pm$0.38) | 76.39 ($\pm$0.38) | 54.75 ($\pm$0.44) |

Figure 9 graphically illustrates the model's performance as a function of the local features available to the active party. An increase in local features positively influences Local-AUC, demonstrating improved model accuracy, while inversely affecting LL-AUC, indicating enhanced privacy protection. The FL-AUC curve remains stable, showcasing the federated learning model's resilience against local feature variability. This figure highlights our method's capability to maintain a balance between accuracy and privacy, aligning with our theoretical foundations.

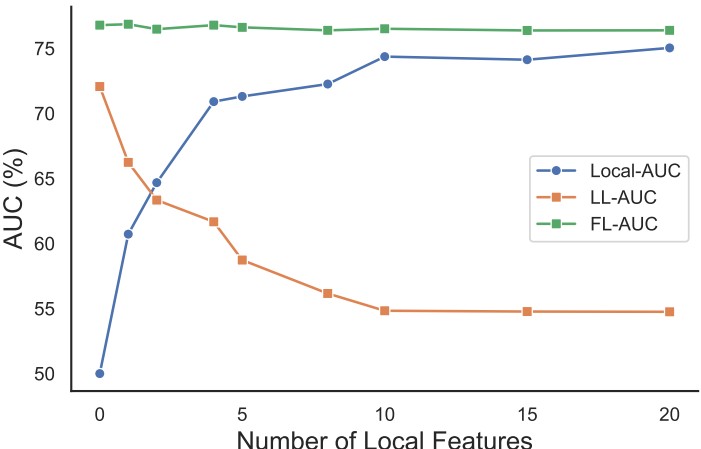

Figure 9: The impact of local features on Criteo dataset performance, emphasizing LPSC's role in enhancing Local-AUC for stronger privacy protection (lower LL-AUC) while maintaining a consistent FL-AUC.

