# OpenReview forum: "Label Privacy Source Coding in Vertical Federated Learning"
_ICLR.cc/2024/Conference — ICLR 2024 Conference Withdrawn Submission_

### Official Review · Reviewer_wj8U · 2023-10-24

**Soundness:** 3 good
**Presentation:** 1 poor
**Contribution:** 3 good
**Rating:** 5
**Confidence:** 4

**Summary:**

This paper studies the problem of protecting label privacy in vertical federated learning. In this setting, training data is vertically split into features and labels. There is one special player named Active party that owns labels and some features; there can be multiple Passive Parties that own disjoint features. Previous works have identified severe privacy issues when applying vanilla vertical federated learning algorithms. This work proposes a novel method of adjusting weight and label of each sample without leaking raw label information. In this paper, the authors make the assumption that the active party has features that are informative of the labels, and use gradient boosting to learn the new label-ID joint distribution. The privacy can be further enhanced with adversarial training. The experiment results show that the proposed method can achieve better privacy-utility trade-offs.

**Strengths:**

The idea of adjusting sample weights are novel and interesting. The closed form solution of AdaBoost is clean and nice. The experiment results are promising, showing that the proposed method is strong against listed attacks.

**Weaknesses:**

My biggest concern is that the setting may be too restricted. In particular, the active party must own informative features. Let us look at Theorem 1. In the very extreme case, assume that the KL-divergence of $p_{gt}$ and $p_{act}$ is 0. Then Theorem 1 says that the LPSC does not leak anything. However, in this case we do not need federated learning: the active party can just learn with its own features and no communication with passive parties are needed. It requires more justification for the setting (active party with informative features, label only privacy, etc) considered in this work.

Mutual information is known to be closely related privacy. [1] is an important reference.

[1] Cuff, Paul, and Lanqing Yu. "Differential privacy as a mutual information constraint." Proceedings of the 2016 ACM SIGSAC Conference on Computer and Communications Security. 2016.

**Questions:**

- It looks like all the datasets are for binary classification. Can LPSC work with non-binary labels?
- According to Eq.5, it seems that in adversarial training, the original labels are used. Will this bring extra privacy issues?
- Is it possible to prove formal privacy guarantees, e.g. differential privacy for LPSC?
- In Appendix B.1, you mentioned " we can use boosting to reduce the bias of the local model by using passive parties’ auxiliary features." I am a bit confused, does LPSC uses information from passive parties?

---

> ### Author Response · Authors · 2023-11-18
> **Weakness 1**
>
> > My biggest concern is that the setting may be too restricted. In particular, the active party must own informative features. Let us look at Theorem 1. In the very extreme case, assume that the KL-divergence of $p_{act}$ and $p_{gt}$ is 0. Then Theorem 1 says that the LPSC does not leak anything. However, in this case, we do not need federated learning: the active party can just learn with its own features, and no communication with passive parties is needed. It requires more justification for the setting (active party with informative features, label-only privacy, etc.) considered in this work.
>
> **Answer**:
>
> We appreciate Reviewer `wj8U`'s concern regarding the VFL setting in our work.
> We emphasize that it is a typical and realistic scenario in many cross-enterprise collaborations, where the active party possesses somewhat/partially informative features. For example, in **cross-company advertisement** [1], an e-commerce company (e.g., Amazon) with user purchase history and a social platform (e.g., Facebook) with user social data collaboratively predict the Click-Through Rate (CTR). Another example is **cross-bank fraud detection** faced by VISA [2], where each bank has some transaction records of a user to produce imprecise local predictions. In both cases, each party holds valuable but partial information, making VFL a viable approach to leverage this marginal information.
>
> The authors acknowledge that the original Theorem 1 misleadingly implied that the active party must own very informative features for LPSC to be effective. To clarify, we have provided a new $\epsilon$-MIP privacy guarantee in the General Response Part 3, as restated as follows for convenience:
>
> **Theorem 1** (Privacy Guarantee.)
> LPSC satisfies $\epsilon$-mutual information privacy ($\epsilon$-MIP). The privacy leakage is bounded by $\epsilon = H(p_{gt}(i,y)|p_{act}(i,y))$, the conditional entropy of the ground-truth label distribution $p_{gt}(i,y)$ given the active party's label distribution $p_{act}(i,y)$. Formally,
> $$
> I(p_{gt}(i,y); p^*_{lpsc}(i,y)) \leq \epsilon \text{ bits},
> $$
> where $p^*_{lpsc}(i,y)$ represents the optimal solution of the LPSC problem.
>
> Addressing the extreme case where $D_{KL}(p_{act}|p_{gt}) = 0$, it's true that VFL may seem unnecessary if the active party can independently learn $p_{gt}$.
> However, our framework's focus, as reflected in the updated privacy guarantee (Theorem 1), is on **ensuring minimal unnecessary privacy leakage** of $p_{act}(i,y)$ even when the active party seeks **marginal** performance improvements through VFL, given $D_{KL}(p_{act}|p_{gt})$ is low.
>
> Lastly, when the active party has no informative features but only labels, the problem setting degrades to another line of research named **Split Learning** [3], where a label-holder and feature-holders train a split neural network. However, in VFL settings [4,5], it is a widely adopted assumption that the active party holds partially informative features.
>
> ------
>
> **References:**
>
> [1] Penghui Wei, Hongjian Dou, Shaoguo Liu, Rongjun Tang, Li Liu, Liang Wang, and Bo Zheng. "FedAds: A Benchmark for Privacy-Preserving CVR Estimation with Vertical Federated Learning." arXiv preprint arXiv:2305.08328 (2023).
>
> [2] https://usa.visa.com/about-visa/visa-research/research-areas.html
>
> [3] Praneeth Vepakomma, Otkrist Gupta, Tristan Swedish, and Ramesh Raskar. "Split learning for health: Distributed deep learning without sharing raw patient data." arXiv preprint arXiv:1812.00564 (2018).
>
> [4] Chen Zhang, Yu Xie, Hang Bai, Bin Yu, Weihong Li, and Yuan Gao. "A survey on federated learning." Knowledge-Based Systems 216 (2021): 106775.
>
> [5] Yang Liu, Yan Kang, Tianyuan Zou, Yanhong Pu, Yuanqin He, Xiaozhou Ye, Ye Ouyang, Ya-Qin Zhang, and Qiang Yang. "Vertical federated learning." arXiv preprint arXiv:2211.12814 (2022).

---

> ### Author Response · Authors · 2023-11-18
> **Weakness 2**
>
> > Mutual information is known to be closely related privacy. [1] is an important reference.
>
> >[1] Paul Cuff, and Lanqing Yu. "Differential privacy as a mutual information constraint." Proceedings of the 2016 ACM SIGSAC Conference on Computer and Communications Security. 2016.
>
> **Answer**:
>
> We appreciate the valuable reference [1] provided by the reviewer to clarify the relationship between mutual information (MIP) adopted by our work and differential privacy (DP).
>
> We provide a general response to the relationship between MIP and DP in the **General Response**.
> We have also added Appendix C.4 "Comparison of MIP v.s. DP" in our revision to clarify the relationship between MIP and DP, based on the mutual information-variant differential privacy, MI-DP, proposed in [1].
>
> For your convenience, we briefly summarize the relationship between MIP and DP as follows:
>
> > **$\epsilon$-MI-DP:**
> > According to [1], a mechanism $\mathcal{M}$ satisfies $\epsilon$-mutual-information differential privacy for some $\epsilon \in \mathbb{R}^{+}$ if, for **any neighboring inputs $X, X'$**, the conditional mutual information between $X$ and $Y=\mathcal{M}(X)$ conditioned on $X'$ satisfies
> > $$
> >        I(X; Y | X') \leq \epsilon \text{ bits}.
> > $$
>
> > **$\epsilon$-MIP:**
> > According to [2], a mechanism $\mathcal{M}$ satisfies $\epsilon$-MIP for some $\epsilon \in \mathbb{R}^{+}$ if, for **any input $X$**, the mutual information between $X$ and the output $Y = \mathcal{M}(X)$ is bounded by $\epsilon$ bits, formally:
> > $$
> >     I(X; Y) \leq \epsilon \text{ bits}.
> > $$
>
> The fundamental difference between MIP and DP, including its variant MI-DP, lies in their underlying **threat models**.
> DP operates under a strong adversary assumption, considering adversaries that have access to neighboring input databases $X'$. In contrast, MIP is designed under the assumption that the adversary lacks prior knowledge of the database, which aligns more closely with the threat model in our VFL setting.
>
> Given the assumption in our VFL setting where an adversary has **no** prior knowledge of the active party's data $\\{\mathbf{i}, \mathbf{y}, \mathbf{X}_0\\}$, MIP emerges as a more fitting choice. Our threat model, as elaborated in Section 3, coincides with the principles of MIP, making it a natural fit for our research.
>
> Meanwhile, [2] reveals a fundamental connection between mutual information and differential privacy bridged by _identifiability_.
>
> We have added the two references [1,2] in our updated version.
>
> ------
>
> **References:**
>
> [1] Paul Cuff, and Lanqing Yu. "Differential privacy as a mutual information constraint." Proceedings of the 2016 ACM SIGSAC Conference on Computer and Communications Security. 2016.
>
> [2] Wang Weina, Lei Ying, and Junshan Zhang. "On the relation between identifiability, differential privacy, and mutual-information privacy." IEEE Transactions on Information Theory 62, no. 9 (2016): 5018-5029.

---

> ### Author Response · Authors · 2023-11-18
> **Question 1 and 2**
>
> ### Question 1
>
> > It looks like all the datasets are for binary classification. Can LPSC work with non-binary labels?
>
> **Answer**:
>
> LPSC is indeed adaptable to non-binary (multi-class) labels, aligning with the multi-class variants of boosting algorithms like AdaBoost and LogitBoost. The transition from binary-class to multi-class settings has been thoroughly explored [1, 2]. As an example, consider the AdaBoost algorithm's extension to a $K$-class problem.
>
> In the binary-class AdaBoost:
> $$
>     p_{lpsc}(i)=\frac{ e^{-\alpha \\;y_i f_{\theta}(i)}}{\sum_{i \in \mathbf{i}} e^{-\alpha \\; y_i f_{\theta}(i)}},
> $$
> where $\alpha=\frac{1}{2}\ln(\frac{1-\epsilon}{\epsilon})$, and $\epsilon$ is the classification error of the local model $f_{\theta}$.
>
> Extended to $K$-classes (SAMME [1]), the equation modifies as:
> $$
>     p_{lpsc}(i)=\frac{ e^{-\alpha \cdot \mathbb{I}(y_i \neq f_{\theta}(i))}}{\sum_{i \in \mathbf{i}} e^{-\alpha \cdot \mathbb{I}(y_i \neq f_{\theta}(i))}},
> $$
> with $\alpha=\frac{1}{2}\ln(\frac{1-\epsilon}{\epsilon}) + \frac{1}{2} \log(K-1)$.
>
> This modification adds a simple term $\log(K-1)$ to $\alpha$. When $K=2$, SAMME simplifies to AdaBoost.
>
> We are actively working to incorporate additional experiments on multi-class classification datasets in our later revised manuscript.
> It's important to note, however, that fully exploring LPSC's extension to multi-class scenarios and assessing privacy-utility trade-offs across different multi-class boosting algorithms is beyond the scope of our current research. This area requires a dedicated and extensive study.
> A promising future research direction could be the development of a tailored multi-class boosting algorithm specifically designed to satisfy LPSC, rather than adapting existing algorithms.
>
> **References:**
>
> [1] Trevor Hastie, Saharon Rosset, Ji Zhu, and Hui Zou. "Multi-class adaboost." Statistics and its Interface 2, no. 3 (2009): 349-360.
>
> [2] Mohammad Saberian, and Nuno Vasconcelos. "Multiclass boosting: Theory and algorithms." Advances in neural information processing systems 24 (2011).
>
>
>
> --------
>
> ### Question 2
>
> > According to Eq.5, it seems that in adversarial training, the original labels are used. Will this bring extra privacy issues?
>
> **Answer:**
>
> Thank you for your pertinent query regarding the use of original labels in our adversarial training process. While these labels are indeed utilized, our methodology is designed to **neutralize or eliminate** any sensitive label information within the LPSC-encoded residuals. The key is the strategic use of the weight $\beta$ in our algorithm, which is meticulously tuned to ensure a harmonious balance between privacy protection and utility enhancement.
>
> The utilization of adversarial loss to protect privacy has recently emerged as an effective strategy for protecting both feature and label privacy in federated learning [1, 2].
> These methods demonstrate how a finely adjusted adversarial loss term that uses **original** private labels or features can mitigate privacy leakage, facilitating a nuanced privacy-utility trade-off without introducing additional privacy risks.
>
> **References:**
>
> [1] Jiankai Sun, Yuanshun Yao, Weihao Gao, Junyuan Xie, and Chong Wang. "Defending against reconstruction attack in vertical federated learning." arXiv preprint arXiv:2107.09898 (2021).
>
> [2] Jingwei Sun, Zhixu Du, Anna Dai, Saleh Baghersalimi, Alireza Amirshahi, David Atienza, and Yiran Chen. "Robust and IP-Protecting Vertical Federated Learning against Unexpected Quitting of Parties." arXiv preprint arXiv:2303.18178 (2023).

---

> ### Author Response · Authors · 2023-11-18
> **Question 3 and 4**
>
> ### Question 3:
>
> > Is it possible to prove formal privacy guarantees, e.g. differential privacy for LPSC?
>
> **Answer**:
>
> We appreciate the reviewer for concerning the privacy guarantee of the proposed privacy mechanism LPSC.
> We provide a formal privacy guarantee with proof that LPSC satisfies $\epsilon$-mutual information privacy ($\epsilon$-MIP) in the **General Response Part 3** and the revised version Appendix B.3 "Privacy Guarantee".
> As discussed in the "MIP v.s. DP" in the **General Response Part 4**, the choice of privacy definition $\epsilon$-MIP is determined by our threat model, which is different from DP.
>
> We restate the $\epsilon$-MIP privacy guarantee for your convenience:
>
> > **Theorem 1** (Privacy Guarantee.)
> > LPSC satisfies $\epsilon$-mutual information privacy ($\epsilon$-MIP). The privacy leakage is bounded by $\epsilon = H(p_{gt}(i,y)|p_{act}(i,y))$, the conditional entropy of the ground-truth label distribution $p_{gt}(i,y)$ given the active party's label distribution $p_{act}(i,y)$. Formally,
> > $$
> >    I(p_{gt}(i,y); p^*_{lpsc}(i,y)) \leq \epsilon \text{ bits},
> > $$
> > where $p^*_{lpsc}(i,y)$ represents the optimal solution of the LPSC problem.
>
> >_Proof._ We approach the LPSC problem, optimizing $p_{lpsc}(i, y)$ with respect to:
> >$$
> >    p^*_{lpsc}(i,y) = \text{arg} \max_{p_{lpsc}(i,y)} I(p_{gt}(i, y); p_{lpsc}(i, y))
> >$$
> >
> >$$
> >    s.t.  \\; I(p_{act}(i, y); p_{lpsc}(i, y)) = 0.
> >$$
> >
> >The key constraint is that $p_{act}(i, y)$ and $p_{lpsc}(i, y)$ must remain independent, which implies that mutual information between \( p_{gt}(i, y) \) and \( p_{lpsc}(i, y) \) excludes any shared information with $p_{act}(i, y)$. Analytically, we express this as:
> >$$
> >    I(p_{gt}(i, y); p^*_{lpsc}(i, y))
> >    =  I(p_{gt}(i, y); p^*_{lpsc}(i, y)|p_{act}(i, y))
> >    \leq  H(p_{gt}(i, y)|p_{act}(i, y))
> >   =  \epsilon \text{ bits},
> >$$
> >where $H(p_{gt}(i, y)|p_{act}(i, y))$ represents the conditional entropy, or the remaining uncertainty in $p_{gt}(i, y)$ after observing $p_{act}(i, y)$.
> >
> >Therefore, the solution $p^*_{lpsc}(i, y)$ satisfies the $\epsilon$-MIP criterion, effectively bounding the mutual information and safeguarding label privacy in accordance with the $\epsilon$-MIP definition.
>
>
> ------
>
>
> ### Question 4
>
> > In Appendix B.1, you mentioned "we can use boosting to reduce the bias of the local model by using passive parties' auxiliary features." I am a bit confused, does LPSC use information from passive parties?
>
> **Answer**:
>
> We apologize for our unclear writing.
> As clarified in the **General Response Part 2** and the Appendix B in the updated version, LPSC is a type of privacy mechanism $M_{lpsc}$ that maps the active party's dataset distribution $p_{gt}(i,y,X_0)$, as defined by $D^{loc} = \\{i^{loc}, y^{loc}, X^{loc}_0\\}$ and generates a new joint distribution:
>
> $$
>  p_{lpsc}(i,y) = M_{lpsc}(p_{gt}(i, y, X_0)).
> $$
>
> This sentence depicts the __entire two-phase VFGBoost__ pipeline including:
> "1) fitting labels $\rightarrow$ 2) computing residuals  $\rightarrow$ 3) fitting residuals".
> However, LPSC only corresponds to the first *offline-phase* with two steps: "1) fitting labels $\rightarrow$ 2) computing residuals", which **only uses information from the active party**.
> Instead, step 3 "fitting residuals" corresponds to the *federated training phase* and uses information from passive parties to fit residuals.
>
> In the updated **Appendix C.1** "Gradient Boosting", we have revised this sentence as follows:
> "Therefore, we can use boosting to reduce the bias of the local model by _training passive parties to fit weighted-residuals._
> The predicted residuals are then added to the active party's local predictions to reduce the bias."

---

> ### Author Response · Authors · 2023-11-22
> **A kind reminder regarding our response**
>
> Dear Reviewer,
>
> As the rebuttal period is approaching its end, we kindly remind you to review our submitted response. Your feedback is essential for finalizing our work. We would greatly appreciate any additional feedback you may have.
>
> Thank you for your attention.
>
> Best regards,
>
> The Authors

---

> ### Comment · Reviewer_wj8U · 2023-11-22
> **Thank you for your rebuttal**
>
> Thank you for your thorough rebuttal, which effectively tackles several of my concerns. I appreciate that the reference to Mutual Information Privacy (MIP) proves beneficial for your work. However, it appears that significant editing is required to seamlessly integrate MIP into your existing framework. Notably, MIP is not formally defined in the main text at present. Recognizing the constraints of time during the rebuttal period, I understand the challenges of extensively polishing the paper. In light of this, I suggest carefully revising the draft to incorporate reviewer comments and considering submission to other top-tier venues.

---

### Official Review · Reviewer_3hcA · 2023-10-29

**Soundness:** 3 good
**Presentation:** 3 good
**Contribution:** 3 good
**Rating:** 6
**Confidence:** 3

**Summary:**

The authors propose an offline-phase data cleansing approach to protect label privacy without compromising utility. Specifically, the idea is to formulate a Label Privacy Source Coding (LPSC) problem to remove the redundant label information in the active party’s features from labels, by assigning each sample a new weight and label (i.e., residual) for federated training.

The authors propose the Vertical Federated Gradient Boosting (VFGBoost) framework to address the LPSC problem with a theoretical guarantee. Moreover, given that LPSC only provides upper-bounded privacy enhancement, VFGBoost further enables a flexible privacy-utility trade-off by incorporating adversarial training during federated training. Experimental results on four real-world datasets substantiate the efficacy of LPSC and the superiority of our VFGBoost framework.

**Strengths:**

+ A unique perspective from the offline phase and present an interesting idea.
+ Provide theoretical guarantee and soundness.
+ Provide comprehensive experiments conducted on four real-world datasets in practical scenarios, such as recommendation and healthcare.

**Weaknesses:**

- Related works need to be improved. Section 2 contains many repeated contents.
- The experimental findings could be more detailed.

**Questions:**

1. The authors redefine privacy and introduce a privacy-utility trade-off. In the related work, the authors also mentioned differential privacy, which has a similar trade-off. Could the authors elaborate on the difference between them?
2. How does adversarial training impact the privacy in your experimental findings? Could the authors explicate the insights/findings?
3. The authors give a comprehensive analysis of newly defined privacy and its leakage. The idea is interesting since it introduces a new perspective on privacy. Actually, I feel a little confused about why Definition 1 and Definition 2 are required. What is the insight/intuition of privacy guarantee? What does the newly defined privacy essentially protect? How do you measure the privacy loss in practice/experiments? Why privacy leakage is defined as mutual information?

---

> ### Author Response · Authors · 2023-11-18
> **Response to Reviewer 3hcA, Weakness 1 and Question 1, 2**
>
> ### Response to Reviewer 3hcA
>
> We sincerely thank you for your insightful comments and concerns regarding privacy issues in our work.
> Your feedback has been invaluable in enhancing the clarity and depth of our analysis in these areas.
> We believe that your queries about privacy issues are thoroughly addressed in our **General Response** and the detailed discussions in **Appendix B: "Threat Model and Privacy Analysis"** of the revised manuscript, where we have rigorously redefined our privacy model, refined our threat model, and provided rigorous privacy guarantees.
>
>
> --------
>
> ### Weakness 1
>
> >Related works need to be improved. Section 2 contains many repeated contents.
>
> **Answer**:
>
> We are grateful to Reviewer 3hcA for highlighting the issue of repetitive content in Section 2. We have carefully revised this section to remove redundancy and improve the flow of information.
>
>
> ----------
>
> ### Question 1
>
> > The authors redefine privacy and introduce a privacy-utility trade-off. In the related work, the authors also mentioned differential privacy, which has a similar trade-off. Could the authors elaborate on the difference between them?
>
> **Answer**:
>
> Thank you for your inquiry about the distinctions between the privacy-utility trade-offs in our LPSC framework and differential privacy (DP). We answer this question from two aspects:
>
> **Comparison of Privacy Definitions:**
> We have refined our manuscript to clarify that our privacy definition aligns with $\epsilon$-Mutual Information Privacy ($\epsilon$-MIP), as detailed in the updated Theorem 1 (**General Response, Part 3**).
> Moreover, we compare MIP and DP in the **General Response, Part 4**.
>
> **Comparison of Privacy-Utility Trade-offs:**
> The VFGBoost framework achieves a privacy-utility balance through a two-phase process: 1) *offline-phase cleansing*, enhancing privacy by removing redundant label information, and 2) *training-phase perturbation*, adjusting the privacy-utility trade-off through controlled learning from perturbed data. This approach efficiently utilizes the redundant label information $p_{act}(i,y)$ to enhance utility without compromising privacy.
>
> In contrast, DP primarily relies on adding noise to labels, a method that does not capitalize on redundant label information for utility improvement. Our framework's nuanced approach, therefore, offers a more tailored balance in VFL settings, as evidenced by the shift in trade-off curves (Figure 5), indicating improved privacy with minimal impact on utility.
>
> ---------
>
> ### Question 2
>
> > How does adversarial training impact the privacy of your experimental findings? Could the authors explicate the insights/findings?
>
> **Answer**:
>
>
> **Impact of Adversarial Training on Privacy:**
> Adversarial training in VFGBoost significantly enhances label privacy by neutralizing the label information in the LPSC-encoded $p_{lpsc}(i,y)$. This is achieved through a negative adversarial loss term in the overall objective, resulting in gradients that effectively eliminate sensitive label information. Consequently, the updated bottom model, trained with these neutralized gradients, preserves label privacy by not learning or retaining critical label knowledge.
>
> **Insights from Experimental Findings:**
> Our experimental results (Figure 5) confirm the effectiveness of this approach. VFGBoost's trade-off curves are positioned closer to the upper-left corner than those of LPSC+Marvell and LPSC+LabelDP, showcasing a better balance between privacy and utility. These findings underscore the advantages of adversarial training in the VFGBoost framework, demonstrating its capability to bolster privacy protection while maintaining utility in practical applications.

---

> ### Author Response · Authors · 2023-11-18
> **Question 3**
>
> We thank Reviewer `3hcA` for the insightful feedback on our privacy definitions. The questions raised have been thoroughly addressed in our **General Response**. Each sub-question related to privacy issues is methodically tackled in this comprehensive response, ensuring a clear understanding of our approach and its implications.
>
>
> --------
>
> ### Q3.1: Definition 1 and Definition 2 are not required.
>
> **Answer**:
>
> Upon reflection and considering the valuable suggestions from Reviewers `3hcA`, `wj8U`, and `WYNH`, we have decided that Definitions 1 and 2 are indeed not essential for our framework.
> Accordingly, we have revised **Section 3** of our manuscript, focusing on rigorously defining the threat model and demonstrating how our LPSC approach aligns with $\epsilon$-mutual information privacy ($\epsilon$-MIP). This change streamlines our presentation and strengthens the theoretical foundation of our work.
>
> --------
>
> ### Q3.2: What is the insight/intuition of privacy guarantee?
>
> **Answer**:
>
> We appreciate your inquiry about the original privacy guarantee's intuition.
> Recognizing the need for a more rigorous approach, we revised our privacy guarantee to align with $\epsilon$-mutual information privacy ($\epsilon$-MIP), as detailed in **General Response Part 3**. This update shifts the focus to conditional entropy, reflecting a more precise measurement of privacy leakage.
>
> **Theorem** (Privacy Guarantee.)
> LPSC satisfies $\epsilon$-mutual information privacy ($\epsilon$-MIP). The privacy leakage is bounded by $\epsilon = H(p_{gt}(i,y)|p_{act}(i,y))$, the conditional entropy of the ground-truth label distribution $p_{gt}(i,y)$ given the active party's label distribution $p_{act}(i,y)$. Formally,
>
> $$
>     I(p_{gt}(i,y); p^*_{lpsc}(i,y)) \leq \epsilon \text{ bits},
> $$
>
> where $p^*_{lpsc}(i,y)$ represents the optimal solution of the LPSC problem.
>
> **Intuition:** The core intuition of the revised guarantee is that privacy leakage in LPSC is inversely related to the amount of label information the active party can infer from its local features. The more label information learned by the active party, the more effective LPSC is in preserving privacy for passive parties' bottom models.
>
>
> ---------
>
>
> ### Q3.3: What does the newly defined privacy essentially protect?
>
> **Answer**:
>
> In our framework, the newly defined privacy mechanism LPSC, as delineated by our privacy guarantee, strategically decouples the private label information $p_{gt}(i,y)$ into two distinct parts.
> It primarily focuses on protecting the label information $p_{act}(i,y)$ that the active party has learned from its local features.
> Meanwhile, LPSC releases the other component, $p_{lpsc}(i,y)$, which is the minimum-sufficient label knowledge to train passive parties.
>
> -------
>
> ### Q3.4: How do you measure the privacy loss in practice/experiments?
>
> **Answer**:
>
> In our experiments, privacy loss is quantified by employing various attack methods targeting the label information derived from the bottom models' forward embeddings.
> This approach aligns with our threat model (referenced in **General Response, Part 1**), where the adversary aims to minimize the expected error using functions $A(\cdot)$ from a set of attacks (e.g., Norm, PMC) as follows:
>
> $$
>     \min_{A \in \mathbb{A}} R_{p_{gt}(i,y)}(A \circ h_{\psi_k}) = \min_{A \in \mathbb{A}} E_{i \sim p_{gt}(i)}[D_{KL}( p_{gt}(y|i) || A(h_{\psi_k}(i)) )],
> $$
>
> Practically, rather than computing the expected KL-divergence directly, we assess the efficacy of these attacks by calculating the Label Leakage AUC score (LL-AUC) from the attack outcomes. Such LL-AUC metric aligns with the adversary's objective in our threat model.
>
>
> --------
>
> ### Q3.5: Why privacy leakage is defined as mutual information?
>
> **Answer**:
>
> As detailed in **General Response Part 2** and restated for convenience, we define privacy leakage as mutual information for specific reasons in our context.
>
> Our rationale for quantifying privacy leakage as mutual information is rooted in the specific adversary's objective in our threat model.
> LPSC involves a privacy mechanism $M_{lpsc}$ that transforms the active party's dataset distribution $p_{gt}(i, y, X_0)$ into a new joint distribution $p_{lpsc}(i,y)$.
> This mechanism is designed to maximize **expected error** of label estimation, which we have determined is equivalent to minimizing the mutual information between $p_{gt}(i,y)$ and $p_{lpsc}(i,y)$. Formally,
>
> $$
>     \max_{p_{lpsc}(i,y)} R_{p_{gt}(i,y)}(p_{lpsc}(i,y)) \iff \min_{p_{lpsc}(i,y)} I(p_{gt}(i,y); p_{lpsc}(i,y)),
> $$
>
> where $I(\cdot; \cdot)$ denotes the mutual information.
>
> Thus, minimizing mutual information effectively enhances label privacy by maximizing the difficulty for an adversary to accurately infer sensitive label information.

---

> ### Author Response · Authors · 2023-11-22
> **A kind reminder regarding our response**
>
> Dear Reviewer,
>
> As the rebuttal period is approaching its end, we kindly remind you to review our submitted response. Your feedback is essential for finalizing our work. We would greatly appreciate any additional feedback you may have.
>
> Thank you for your attention.
>
> Best regards,
>
> The Authors

---

> > ### Comment · Reviewer_3hcA · 2023-11-22
> >
> > Thanks for the response.
> > I would like to keep my rating.

---

### Official Review · Reviewer_pUxt · 2023-10-31

**Soundness:** 3 good
**Presentation:** 2 fair
**Contribution:** 3 good
**Rating:** 5
**Confidence:** 5

**Summary:**

This paper proposed a novel framework to defend the data leakage problems in VFL, an important issue in collaborative learning scenarios.   The proposed LPSC approach is based on  mutual information optimization, which is solved by boosting approaches, and is able to reduce label leakage while preserving model utility in the evaluated experimental settings. Combination of LPSC and other methods also demonstrate promising utility-privacy trade-off.

**Strengths:**

1. The paper proposed a novel approach to defend label attacks in VFL settings and demonstrate experimentally the effectiveness of the method on multiple datasets.

2. Theoretical analysis is conducted to guide the design of the framework.

3. The proposed approach is able to achieve privacy protection without hurting model utility.

**Weaknesses:**

1. Notations are inconsistent and confusing, making the paper hard to follow. There are many inconsistent notations such as X_0^loc and X_0, y^loc and y, i^loc and i, both denoting active party's data, p*_{lpsc} and p_{lpsc} etc. In Figure 3, it is unclear whether it illustrates the joint distribution or weight distribution. Eq.4 is also confusing, in which the federated model training is a function of i, the ID. The authors are suggested to proofread the entire manuscript to make notations easy to understand and consistent.

2. Experimental evaluations are insufficient to support the main claims of the work. The comparison with other methods are only conducted  on PMC attack, when the adversarial loss of the proposed method is trained also against PMC attack. No comparison on other label leakage attacks or feature attacks are provided, whereas feature leakage attacks are also considered in the security definition.

3. It appears that the effectiveness of proposed method depends on the quality of the local features of the active party. In the extreme case that no features are available on the active party but labels, the method may not apply. The impact of the importance of the local features of the active party needs to be evaluated.

4. Some implementation details are missing. For example, how is LPSC+DP or LPSC+Marvell trained? why not LPSC+MID?

**Questions:**

1. How are importance of the local features affect the evaluation?

2. How are the proposed method compared with other approaches on attacks other than PMC?

3. Can you explain why it is reasonable to assume the conditional distribution to be the same in Theorem 2?

4.how is LPSC+DP or LPSC+Marvell trained? why not LPSC+MID?

---

> ### Author Response · Authors · 2023-11-18
> **Weakness 1**
>
> > Notations are inconsistent and confusing, making the paper hard to follow. There are many inconsistent notations such as $X_0^{loc}$ and $X_0$, $y^{loc}$ and $y$, $i^{loc}$ and $i$, both denoting active party's data, $p^*_{lpsc}$ and $p_{lpsc}$ etc. In Figure 3, it is unclear whether it illustrates the joint distribution or weight distribution. Eq.4 is also confusing, in which the federated model training is a function of $i$, the ID. The authors are suggested to proofread the entire manuscript to make notations easy to understand and consistent.
>
> **Answer**:
>
> We sincerely apologize for any confusion caused by the unclear writing and appreciate your valuable feedback. To clarify, the notations used are consistent and carefully defined in __Section 3__, Subsection *Vertical Federated Learning Setting*. We understand that the complexity of the subject might lead to some initial confusion, and we have conducted a thorough review and update of the manuscript to improve clarity and consistency in notations.
>
> 1. **Local Data Notations**: We differentiate between aligned data $\mathcal{D} = \\{\mathbf{i}, \mathbf{y}, \mathbf{X}_0, \ldots, \mathbf{X}_K \\}$ and local data $\mathcal{D}^{loc} = \\{\mathbf{i}^{loc}, \mathbf{y}^{loc}, \mathbf{X}_0^{loc}\\}$, which includes both aligned samples in $\mathcal{D}$ and extra unaligned local samples. We have clarified this in Figure 1(a) in the revised version.
>
> 2. **Simplified Notations for Functions**: We use sample ID $i$ as a concise notation to represent feature vectors, such as $f_{\theta}(i)$ for $f_{\theta}(x_{0,i})$ and $h_{\psi_k}(i)$ for $h_{\psi_k}(x_{k,i})$. We restated the simplified notations in Equation (4) for clarity.
>
> 3. **Optimal LPSC Solution**: The notation $p^*_{lpsc}$ represents the optimal solution to the LPSC problem, fulfilling specific criteria for mutual information. In Figure 3, we have revised the notations $p_{gt}, p_{act}, p^*_{lpsc}$ to clearly indicate $p_{gt}(i,y), p_{act}(i,y), p^*_{lpsc}(i, y)$, denoting $p^*_{lpsc}(i, y)$ represents the optimal joint distribution.
>
> We extend our gratitude to the reviewer for this insightful feedback. Our revisions aim to not only rectify the noted issues but also to significantly improve the overall readability and understanding of our manuscript.

---

> ### Author Response · Authors · 2023-11-18
> **Weakness 2, the same as Question 2**
>
> > How are the proposed method compared with other approaches on attacks other than PMC? Experimental evaluations are insufficient to support the main claims of the work. The comparison with other methods is only conducted on PMC attack, when the adversarial loss of the proposed method is trained also against PMC attack. No comparison on other label leakage attacks or feature attacks are provided, whereas feature leakage attacks are also considered in the security definition.
>
> **Answer**:
>
> > **Weakness 2.1: "Comparisons against PMC attack are insufficient to support the main claims of the work."**
>
> We gratefully acknowledge your valuable feedback on the scope of our evaluations in Section 5.3.
>
> We would like to humbly clarify that the core focus of our research revolves around the **LPSC** mechanism, primarily aimed at bolstering label privacy in VFL through a novel two-phase process: 1) offline-phase cleansing and 2) training-phase perturbation. The experiments are designed to support the main claim that **LPSC-based two-phase framework is superior in enhancing label privacy in VFL**. While VFGBoost (i.e., LPSC+Adv. Training) is just one among several implementations of LPSC-based two-phase methods, including LPSC+DP and LPSC+Marvell.
>
> LPSC's effectiveness against various attacks is validated in Section 5.2. Specifically, in Section 5.3, the integration of LPSC in three perturbation methods (adversarial training, LabelDP, and Marvell) consistently outperforms baselines, validating LPSC's effectiveness. In other words, _even if replacing adversarial training in VFGBoost with other perturbations, the LPSC-based two-phase framework still outperforms baselines_.
>
> Moreover, the PMC attack is particularly threatening to our setting for two key reasons:
>
> 1. **Prior knowledge.** The PMC attack assumes the adversary knows some labeled data **prior**, which is a strong adversary assumption. This contrasts with other attacks like Norm and Spectral, which do not assume prior knowledge, leading to heuristic attack functions. The superiority of the PMC attack is also confirmed in our experimental results in _Table 2_.
>
> 2. **Tailored threat model.** PMC specifically targets label inference from forward embeddings, aligning closely with our focus on label privacy in VFL settings. While the other attacks were initially designed to attack labels from **gradients**, they are adopted to fit our setting.
>
> In summary, while PMC was a primary focus due to its relevance and stringency, we acknowledge the value in broader attack comparisons and are actively exploring this in ongoing research.
>
> **Weakness 2.2: Feature Attack**:
>
> Regarding feature privacy, we have evaluated the protection against the Model Inversion attack in **Appendix G.3**. While not our primary focus, LPSC indirectly enhances feature privacy by enabling the use of lower-dimensional embeddings, thus reducing potential feature information leakage.

---

> > ### Author Response · Authors · 2023-11-21
> > **Weakness 3, the same as Question 1**
> >
> > > Weakness 3: It appears that the effectiveness of the proposed method depends on the quality of the local features of the active party. In the extreme case that no features are available on the active party but labels, the method may not apply. The impact of the importance of the local features of the active party needs to be evaluated.
> > >
> > > Question 1: How does the importance of the local features affect the evaluation?
> >
> > **Answer:**
> >
> > We sincerely appreciate your insightful concern regarding the dependency of LPSC's effectiveness on the quality of local features in the active party.
> > In response, we have conducted additional experiments on the Criteo dataset (included in **Appendix G.5**) to thoroughly address this issue.
> >
> > **1. Additional experiments**: According to the results in the following table, we observe that when the active party possesses a minimal feature set, it correlates with a lower Local-AUC and a higher Label Leakage AUC (LL-AUC).
> > Conversely, as the number of local features increases, the Local-AUC improves, which leads to a reduction in LL-AUC, thereby indicating stronger privacy protection.
> > Notably, the Federated model's AUC (FL-AUC) remains relatively stable across these variations, demonstrating the robustness of LPSC.
> > The empirical results are consistent with the privacy guarantee in Theorem 1, where the LPSC satisfies $\epsilon$-MIP bounded by the conditional entropy.
> >
> >
> > | # Local Features | Loc-AUC ($\pm$std) | FL-AUC ($\pm$std) | LL-AUC ($\pm$std) |
> > |------------------|-------------------|-------------------|-------------------|
> > | 0                | 0.5               | 76.79 ($\pm$0.34) | 72.07 ($\pm$0.33) |
> > | 1                | 60.72 ($\pm$0.39) | 76.86 ($\pm$0.28) | 66.24 ($\pm$0.34) |
> > | 2                | 64.68 ($\pm$0.47) | 76.48 ($\pm$0.32) | 63.34 ($\pm$0.47) |
> > | 4                | 70.91 ($\pm$0.48) | 76.79 ($\pm$0.38) | 61.67 ($\pm$0.48) |
> > | 5                | 71.31 ($\pm$0.34) | 76.62 ($\pm$0.24) | 58.73 ($\pm$0.42) |
> > | 8                | 72.26 ($\pm$0.38) | 76.39 ($\pm$0.27) | 56.16 ($\pm$0.48) |
> > | 10               | 74.37 ($\pm$0.35) | 76.51 ($\pm$0.34) | 54.83 ($\pm$0.51) |
> > | 15               | 74.13 ($\pm$0.52) | 76.38 ($\pm$0.43) | 54.77 ($\pm$0.38) |
> > | 20               | 75.04 ($\pm$0.38) | 76.39 ($\pm$0.38) | 54.75 ($\pm$0.44) |
> >
> > **2. Extreme case**: When the active party has no feature but labels, our setting deviates from standard VFL and resembles **Split Learning** [1]. Split Learning involves a label-holder and feature-holders training a network in unison, distinct from VFL where the active party typically holds some informative features.
> > As highlighted in [2, 3], standard VFL aims to marginally enhance performance using contributions from passive parties.
> >
> > We believe these additions and clarifications significantly enhance our manuscript and address the concerns raised.
> > We hope this response and the corresponding updates in the manuscript satisfactorily clarify the issue.
> >
> > -------
> >
> > **References:**
> >
> > [1] Praneeth Vepakomma, Otkrist Gupta, Tristan Swedish, and Ramesh Raskar. "Split learning for health: Distributed deep learning without sharing raw patient data." arXiv preprint arXiv:1812.00564 (2018).
> >
> > [2] Chen Zhang, Yu Xie, Hang Bai, Bin Yu, Weihong Li, and Yuan Gao. "A survey on federated learning." Knowledge-Based Systems 216 (2021): 106775.
> >
> > [3] Yang Liu, Yan Kang, Tianyuan Zou, Yanhong Pu, Yuanqin He, Xiaozhou Ye, Ye Ouyang, Ya-Qin Zhang, and Qiang Yang. "Vertical federated learning." arXiv preprint arXiv:2211.12814 (2022).

---

> > ### Comment · Reviewer_pUxt · 2023-11-22
> >
> > I have read the authors' responses, which addressed some of my concerns. The experimental evaluations are still limited, as comparisons against other baselines are still only performed on one attack whereas various label and feature attacks have been proposed in VFL setting in recent years. I think the paper will be much stronger if it demonstrates its superiority over other methods on various attacks.  The writing of the final draft has been improved overall, but is still notation heavy and not easy to follow. It also introduced new questions and confusions. For example, it appears that there are unaligned data samples which result in the different notations in X^loc and X. However the significance or reason of this assumed setting is not explained. It is also not clear in the experiments how these unaligned samples are set. In summary, I think this paper still needs improvement in experimental evaluations and presentation.

---

> ### Author Response · Authors · 2023-11-18
> **Weakness 4 and Question 3**
>
> ### Weakness 4, the same as Question 4
>
> > Weakness 4: Some implementation details are missing. For example, how is LPSC+DP or LPSC+Marvell trained? Why not LPSC+MID?
> >
> > Question 3: How is LPSC+DP or LPSC+Marvell trained? Why not LPSC+MID?
>
> **Answer**:
>
> We appreciate your query on the implementation details of LPSC+LabelDP and LPSC+Marvell. Both methods and our VFGBoost are grounded in our key insight that label privacy protection in VFL should be a two-phase process: 1) **offline-phase cleansing** through **LPSC**, enhancing privacy without utility compromise, and 2) **training-phase perturbation** for further privacy-utility balancing through various perturbation methods (e.g., Adv. training, LabelDP, Marvell, etc.)
>
> For LPSC+LabelDP and LPSC+Marvell, the active party first conducts offline-phase cleansing via LPSC, then proceeds to federated training with LPSC-encoded results. LPSC+LabelDP uses the Multi-stage Training algorithm in [1] with modified sample weights and labels, while LPSC+Marvell employs the Marvell algorithm [3] with LPSC-encoded weights.
>
> Regarding LPSC+MID, the MID [3] approach **inherently integrates cleansing and perturbation** in one step during training, using a mutual information regularization term. This makes MID a parallel baseline to the combined approaches like AdaBoost (i.e., LPSC+adversarial training), LPSC+DP, and LPSC+Marvell, rather than a complementary method that can be combined with LPSC.
>
> **References**:
>
> [1] Badih Ghazi, Noah Golowich, Ravi Kumar, Pasin Manurangsi, and Chiyuan Zhang. "Deep learning with label differential privacy." Advances in neural information processing systems 34 (2021): 27131-27145.
>
> [2] Oscar Li, Jiankai Sun, Xin Yang, Weihao Gao, Hongyi Zhang, Junyuan Xie, Virginia Smith, and Chong Wang. "Label leakage and protection in two-party split learning." arXiv preprint arXiv:2102.08504 (2021).
>
> [3] Tianyuan Zou, Yang Liu, and Ya-Qin Zhang. "Mutual Information Regularization for Vertical Federated Learning." arXiv preprint arXiv:2301.01142 (2023).
>
>
> -------
>
> ### Question 3
>
> > Can you explain why it is reasonable to assume the conditional distribution to be the same in Theorem 2?
>
> **Answer**:
>
> We appreciate your query regarding the assumption of a constant conditional distribution in Theorem 2. This assumption simplifies the LPSC problem to align with AdaBoost for theoretical analysis, but it is **not a strict requirement for all scenarios**, such as those involving LogitBoost and L2-Boost.
>
> When optimizing both the marginal (weights) and conditional (residuals) distributions, the LPSC problem aligns with LogitBoost, as demonstrated in our updated version with the following formulation:
>
> $$
>     p_{lpsc}(i) = \frac{f_{\theta}(i)(1 - f_{\theta}(i))}{\sum_{j \in \mathbf{i}} f_{\theta}(j)(1 - f_{\theta}(j))}, \\; \\;
>     p_{lpsc}(y|i) = \frac{y_i - f_{\theta}(i)}{f_{\theta}(i)(1-f_{\theta}(i))}.
> $$
>
> Our experimental results (Table 3, Section 5.4) further confirm that optimizing both components in LogitBoost yields superior privacy protection compared to AdaBoost, which holds the conditional distribution constant.

---

> ### Author Response · Authors · 2023-11-22
> **A kind reminder regarding our response**
>
> Dear Reviewer,
>
> As the rebuttal period is approaching its end, we kindly remind you to review our submitted response. Your feedback is essential for finalizing our work. We would greatly appreciate any additional feedback you may have.
>
> Thank you for your attention.
>
> Best regards,
>
> The Authors

---

### Official Review · Reviewer_WYNH · 2023-11-02

**Soundness:** 2 fair
**Presentation:** 3 good
**Contribution:** 2 fair
**Rating:** 5
**Confidence:** 4

**Summary:**

This paper presents an approach to enhancing label privacy in vertical federated learning (VFL). The paper first introduces a privacy notion based on the concept of mutual information, then formalizes the privacy protection task into the Label Privacy Source Coding (LPCS) problem, and demonstrates that gradient boosting is a suitable method for solving the LPCS problem. Subsequently, the Vertical Federated Gradient Boosting framework is proposed to efficiently optimize the LPCS problem.

**Strengths:**

1. The proposed solution is interesting.

2. The experimental evaluation is extensive.

**Weaknesses:**

1. The new privacy definition needs to be more formally and rigorously defined. The adversary model and the privacy guarantee provided by the new privacy definition should be clearly delineated. Without a clear description of the adversary's capabilities, the privacy definition's theoretical foundations remain unclear.

2. The paper does not provide a formal analysis of the new privacy definition's properties. In particular, it remains unclear whether this definition adheres to the axioms laid out in https://www.cse.psu.edu/~duk17/papers/axioms.pdf. The paper would be greatly strengthened by a discussion on this aspect, identifying any axioms that are not met and providing a reasoned argument for why such deviations are acceptable within the context of the problem being tackled.

3. An in-depth discussion of why VFGBoost surpasses label DP would be beneficial.

**Questions:**

See the weaknesses

---

> ### Author Response · Authors · 2023-11-18
> **Weakness 1**
>
> > The new _privacy definition_ needs to be more formally and rigorously defined.
> The _adversary model_ and the _privacy guarantee_ provided by the new privacy definition should be clearly delineated.
> Without a clear description of the adversary's capabilities, the privacy definition's theoretical foundations remain unclear.
>
> __Answer__:
>
> Thank you for your insightful comments regarding the need for a more formal and rigorous definition of our privacy model. Your feedback has been instrumental in enhancing the clarity and depth of our manuscript. In response to your concerns, as well as those echoed by other reviewers, we have comprehensively addressed the threat model, privacy definition, and privacy guarantee in the **General Response**.
>
> 1. **Threat Model:**
> For a detailed description of the threat model, which forms the cornerstone of our privacy approach, please refer to **General Response, Part 1**.
>
> 2. **Privacy Definition:**
> We acknowledge the previous ambiguity in our terminology and have rectified this in the revised manuscript. The term "private label information," previously referred to as the privacy definition, is now clearly defined as $p_{gt}(i,y)$ to prevent any confusion. This clarification is elaborated upon in **General Response, Part 2**. Our choice of privacy definition, $\epsilon$-MIP, is directly informed by our threat model and is rigorously defined and justified within our framework.
>
> 3. **Privacy Guarantee:**
> Furthermore, we have substantiated our approach with a formal privacy guarantee, demonstrating that our LPSC mechanism satisfies $\epsilon$-MIP, as detailed in **General Response, Part 3**.
>
> Accordingly, we have revised **Section 3** of our manuscript to provide a clearer exposition of the threat model and privacy definition. We have also introduced **Appendix B: "Privacy Threats and Analysis"** which offers an in-depth exploration of these concepts, including the threat model (B.1), the formal definition of $\epsilon$-Mutual Information Privacy ($\epsilon$-MIP) as our privacy definition (B.2), a detailed privacy guarantee with proof (B.3), and a comparative analysis of MIP and Differential Privacy (DP) (B.4).

---

> ### Author Response · Authors · 2023-11-18
> **Weakness 2**
>
> >  The paper does not provide a formal analysis of the new privacy definition's properties.
> In particular, it remains unclear whether this definition adheres to the axioms laid out in [1].
> The paper would be greatly strengthened by a discussion on this aspect, identifying any axioms that are not met and providing a reasoned argument for why such deviations are acceptable within the context of the problem being tackled.
>
> >[1] Daniel Kifer, and Bing-Rong Lin. "An axiomatic view of statistical privacy and utility." Journal of Privacy and Confidentiality 4, no. 1 (2012).
>
> **Answer**:
>
> We are grateful to receive the valuable suggestion to examine our privacy definition against the axioms outlined in [1]. This has enabled us to further solidify the theoretical underpinnings of our work.
>
> Recall that, in response to Weakness 1, we have clarified that $p_{gt}(i,y)$ is now referred to as **private label information** in our revised manuscript, enhancing clarity.
>
> Our Label Privacy Source Coding (LPSC) corresponds to a category of privacy mechanism that satisfies the privacy definition of **$\epsilon$-Mutual Information Privacy ($\epsilon$-MIP)**, as proved in General Response Part 3. As per the perspective in [1], a privacy definition encompasses a set of privacy mechanisms. Thus, LPSC, with its underlying boosting algorithms (such as AdaBoost, and LogitBoost), can be seen as a privacy definition with specific mechanisms for implementation.
>
> Upon reviewing the axioms in [1], we confirm that LPSC satisfies the following axioms:
>
> **Axiom 1: Transformation Invariance**. This axiom posits that the privacy of sanitized data is maintained through post-processing, provided no sensitive information is directly utilized. __LPSC complies with Axiom 1__ as the post-processing in our approach (specifically, the federated training phase in Algorithm 1, lines 3-9) exclusively utilizes LPSC-encoded residuals and does not directly access ground-truth label information. Consequently, adversarial passive parties cannot glean additional label privacy during federated training beyond what is revealed through LPSC-encoded results.
>
> **Axiom 2: Convexity**. This axiom suggests that the choice of privacy mechanism should be independent of the actual input data. Thereby, a privacy mechanism that randomly selects between two privacy mechanisms also satisfies the privacy definition.
> __LPSC adheres to Axiom 2__ because the selection of specific boosting algorithms (e.g., AdaBoost or LogitBoost) as privacy mechanisms is not influenced by the input data. The choice between these algorithms is arbitrary and does not depend on the nature of the data being processed, thus satisfying the convexity criterion.
>
> -------
>
> **Reference**:
>
> [1] Daniel Kifer, and Bing-Rong Lin. "An axiomatic view of statistical privacy and utility." Journal of Privacy and Confidentiality 4, no. 1 (2012).

---

> ### Author Response · Authors · 2023-11-18
> **Weakness 3**
>
> > An in-depth discussion of why VFGBoost surpasses label DP would be beneficial.
>
> **Answer**:
>
> Thank you for the opportunity to discuss the advantages of our VFGBoost framework over LabelDP [1] in the context of VFL. While LabelDP provides a robust solution for **centralized deep learning** settings by partitioning the centralized training data into multiple subsets and adding noise to labels in multiple stages, it doesn't cater to the unique characteristics of VFL. Specifically, algorithms proposed in LabelDP [1] do not explicitly address scenarios where an active party possesses labeled local data, and passive parties have auxiliary features without labels.
>
> In contrast, VFGBoost is designed explicitly for VFL. It utilizes Label Privacy Source Coding (LPSC) to optimally leverage this unique data distribution structure. By first computing and releasing LPSC-encoded weighted residuals, VFGBoost ensures efficient and tailored label privacy protection. Additionally, it trains the model on perturbed datasets, allowing for a flexible balance between privacy and utility. This specific alignment with VFL's requirements, combined with our framework's practical efficiency and adaptability, positions VFGBoost as a superior alternative to LabelDP in VFL environments.
>
> --------
>
> **Reference**:
>
> [1] Badih Ghazi, Noah Golowich, Ravi Kumar, Pasin Manurangsi, and Chiyuan Zhang. "Deep learning with label differential privacy." Advances in neural information processing systems 34 (2021): 27131-27145.

---

> ### Author Response · Authors · 2023-11-22
> **A kind reminder regarding our response**
>
> Dear Reviewer,
>
> As the ICLR discussion period nears its conclusion, we kindly remind you to review our submitted response. Your feedback is essential for finalizing our work. We would greatly appreciate any additional feedback you may have.
>
> Thank you for your attention.
>
> Best regards,
>
> The Authors

---

### Author Response · Authors · 2023-11-18
**General Response on Threat Model and Privacy Analysis**

We are grateful for the insightful comments from the reviewers regarding the threat model and privacy definition. These constructive comments have significantly contributed to solidifying the theoretical foundations of the proposed LPSC.

This general response is structured to address the four generally concerned issues by three reviewers (WYNH, 3hcA, and wj8U) that have emerged as central to understanding our approach:

- **Part 1: Formal and Rigorous Definition of the Threat Model** - Here, we delineate the threat model employed in our study, explicating its scope and implications for label privacy.
- **Part 2: Private Label Information and Privacy Definition** - This part focuses on why we consider the ID-label joint distribution $p_{gt}(i, y)$ as critical private label information and elaborates on the privacy definition $\epsilon$-Mutual Information Privacy ($\epsilon$-MIP) adopted in our research.
- **Part 3: Privacy Guarantee -- LPSC Satisfies $\epsilon$-MIP** - We prove that LPSC satisfies $\epsilon$-MIP.
- **Part 4: Choice of Mutual Information Privacy (MIP) Over Differential Privacy (DP)** - In this part, we justify our proposed LPSC aligns with MIP over DP, clarifying the differences and relative advantages of MIP in the context of our work.

Corresponding to this general response, we have modified **Section 3** and included a comprehensive section, "Threat Model and Privacy Analysis," in **Appendix B** of the revised manuscript, which offers a detailed analysis of the above-mentioned four parts.

---

> ### Author Response · Authors · 2023-11-18
> **Part 1: Label Privacy Threat Model**
>
> **Adversary's capabilities**: A semi-honest non-colluding passive party $P_k$ has a bottom model $h_{\psi_k}(\cdot)$ which outputs forward embeddings $h_{\psi_k}(i) := h_{\psi_k}(\mathbf{x}_{k,i})$ of the $i$-th sample in its features $\mathbf{X}_k$. However, $P_k$ has *no* prior knowledge of the active party's data {$\mathbf{i}, \mathbf{y}, \mathbf{X}_0$}.
>
> **Adversary's objective**: The adversary aims to minimize the __expected error__ $R_{ p_{gt}(i,y)}$ of label estimation on a dataset with joint ID-label distribution $p_{gt}(i,y)$ by optimizing an attack function $A(\cdot) \in \mathbb{A}$:
>
> $$\min_{A \in \mathbb{A}}   R_{ p_{gt}(i,y)} \left(A \circ h_{\psi_k}  \right) = \min_{A \in \mathbb{A}} E_{i \sim p_{gt}(i)} \left[D_{KL}( p_{gt}(y|i) || A(h_{\psi_k}(i)) ) \right]. $$
>
> The attack function $A(\cdot)$ is determined by the attack methods (e.g., Norm, PMC, etc.)
> We summarize the threat model of label privacy in our setting in the following table:
>
> | Adversary       | Passive party $P_k$                                  |
> |-----------------|------------------------------------------------------|
> | Threat model    | Semi-honest, non-colluding                           |
> | Adversary's knowledge | Forward embeddings $h_{\psi_k}(i) := h_{\psi_k}(x_{k,i})$ from bottom model $h_{\psi_k}(\cdot)$. No prior knowledge of $p_{gt}(i, y, X_0)$, as defined by {$\mathbf{i}, \mathbf{y}, \mathbf{X}_0$}. |
> | Adversary's objective | Minimize expected error $R_{p\_{gt}(i,y)}$ by optimizing attack function $A(\cdot)$. |

---

> ### Author Response · Authors · 2023-11-18
> **Part 2: Private Label Information and Privacy Definition**
>
> Given the above threat model, we now explain __how the threat model naturally leads to the adoption of MIP, instead of DP, as our privacy definition__.
>
> The adversary's goal, as defined in the above equation, is to minimize the *expected error* regarding the ground-truth joint distribution $p_{gt}(i,y)$. Given the adversary has *no* prior knowledge about $p_{gt}(i,y)$ and access only to forward embeddings, the **private label information** *is and only is* the *ID-label joint distribution* $p_{gt}(i, y)$.
>
> To understand how our approach fits within the broader landscape of privacy definitions and mechanisms, we first recall the definition of *privacy definition* and *privacy mechanism* in [1], as mentioned by `Reviewer HYNH`:
>
> > **Definition** (Privacy Definition [1]).
> >    Given an input space $\mathbb{I}$, a privacy definition is a set of randomized algorithms with common input space $\mathbb{I}$. We say that these randomized algorithms satisfy the privacy definition.
>
> >**Definition** (Privacy Mechanism [1]).
> >    A privacy mechanism $M$ is a randomized algorithm that satisfies a privacy definition.
>
> Our proposed privacy definition LPSC corresponds to a category of privacy mechanisms $M_{lpsc} $
> that inputs the active party's dataset distribution $p_{gt}(i, y, X_0)$, and generates a new ID-label joint distribution:
> $$
> p_{lpsc}(i,y) = M_{lpsc} (p_{gt}(i, y, X_0)).
> $$
>
> During federated training, the bottom model $h_{\psi_k}$ is trained to align with $p_{lpsc}(i,y)$. Consequently, the optimal attack error can be expressed as:
>
> $$
> R_{p_{gt}(i,y) } \left(p_{lpsc}(i,y) \right) = \min_{A\in \mathbb{A}} E_{i \sim p_{gt}(i)} \left[  D_{KL}( p_{gt}(y|i) \\; || \\; A(h_{\psi^*_k}(i)) ) \right],
> $$
>
> where the optimal bottom model $\psi^*_k$ is trained as follows:
>
> $$
> \min_{\psi_k} E_{i \sim p_{lpsc}(i,y)} \left[ D_{KL}(p_{lpsc}(y|i) \\; || \\; g(h_{\psi_k}(i) ) ) \right].
> $$
>
> To enhance label privacy, the privacy mechanism $M_{lpsc}$ should be designed to maximize this attack error $R_{p_{gt}(i,y)}(p_{lpsc}(i,y) )$. We observe that maximizing $R_{p_{gt}(i,y)}(p_{lpsc}(i,y) )$ is equivalent to minimizing the mutual information between $p_{gt}(i,y)$ and $p_{lpsc}(i,y)$, i.e.,
>
> $$
> \max_{p_{lpsc}(i,y)} R_{p_{gt}(i,y)}(p_{lpsc}(i,y)) \iff \min_{p_{lpsc}(i,y)} I(p_{gt}(i,y); p_{lpsc}(i,y)),
> $$
>
> where $I(\cdot; \cdot)$ denotes the mutual information.
> Thus, a privacy mechanism $M_{lpsc}$ minimizing mutual information $I(p_{gt}(i,y);p_{lpsc}(i,y))$ equivalently maximizes the attack error, protecting label privacy.
> Accordingly, **our threat model aligns with the principle of $\epsilon$-mutual information privacy ($\epsilon$-MIP) [2]**:
>
> >**Definition**($\epsilon$-Mutual Information Privacy [2]).
> >    A mechanism $\mathcal{M}$ satisfies $\epsilon$-MIP for some $\epsilon \in \mathbb{R}^{+}$ if, for any input $X$, the mutual information between $X$ and the output $Y = \mathcal{M}(X)$ is bounded by $\epsilon$ bits, formally:
> >    $$
> >        I(X; Y) \leq \epsilon \text{ bits}.
> >    $$
>
> -------
>
> **References**
>
> [1] Daniel Kifer, and Bing-Rong Lin. "An axiomatic view of statistical privacy and utility." Journal of Privacy and Confidentiality 4, no. 1 (2012).
>
> [2] Wang Weina, Lei Ying, and Junshan Zhang. "On the relation between identifiability, differential privacy, and mutual-information privacy." IEEE Transactions on Information Theory 62, no. 9 (2016): 5018-5029.

---

> ### Author Response · Authors · 2023-11-18
> **Part 3: Privacy Guarantee**
>
> We prove that our proposed __LPSC satisfies the $\epsilon$-MIP__.
>
> **Theorem 1** (Privacy Guarantee.)
> *LPSC satisfies $\epsilon$-mutual information privacy ($\epsilon$-MIP). The privacy leakage is bounded by $\epsilon = H(p_{gt}(i,y)|p_{act}(i,y))$, the conditional entropy of the ground-truth label distribution $p_{gt}(i,y)$ given the active party's label distribution $p_{act}(i,y)$. Formally,*
> $$
> I(p_{gt}(i,y); p^*_{lpsc}(i,y)) \leq \epsilon \text{ bits},
> $$
> *where $p^{\*}_{lpsc}(i,y)$ represents the optimal solution of the LPSC problem.*
>
> -----
>
> *Proof:*
> We approach the LPSC problem, optimizing $p_{lpsc}(i, y)$ with respect to:
> $$
> p^*_{lpsc}(i,y) = \text{arg} \max_{p_{lpsc}(i,y)} I(p_{gt}(i, y); p_{lpsc}(i, y))
> $$
>
> $$
> \text{s.t.} \\; I(p_{act}(i, y); p_{lpsc}(i, y)) = 0.
> $$
>
> The key constraint is that $p_{act}(i, y)$ and $p_{lpsc}(i, y)$ must remain independent, which implies that mutual information between $p_{gt}(i, y)$ and $p_{lpsc}(i, y)$ excludes any shared information with $p_{act}(i, y)$. Analytically, we express this as:
> $$
> \begin{align*}
> & I(p_{gt}(i, y); p^*_{lpsc}(i, y)) \\
> = & I(p_{gt}(i, y); p^*_{lpsc}(i, y)|p_{act}(i, y)) \\
> \leq & H(p_{gt}(i, y)|p_{act}(i, y)) \\
> = & \epsilon \text{ bits},
> \end{align*}
> $$
> where $H(p_{gt}(i, y)|p_{act}(i, y))$ represents the conditional entropy, or the remaining uncertainty in $p_{gt}(i, y)$ after observing $p_{act}(i, y)$.
>
> Therefore, the solution $p^*_{lpsc}(i, y)$ satisfies the $\epsilon$-MIP criterion, effectively bounding the mutual information and safeguarding label privacy in accordance with the $\epsilon$-MIP definition.

---

> ### Author Response · Authors · 2023-11-18
> **Part 4: Comparison of MIP v.s. DP**
>
> In this part, we justify our proposed LPSC aligns with MIP over DP, clarifying the differences and relative advantages of MIP in the context of our work.
>
> Differential privacy (DP) ensures that the output of a mechanism does not significantly change with the alteration of a single record in the dataset. In contrast, mutual information privacy (MIP), which we adopt in this paper, focuses on limiting the mutual information between the input and output of a privacy mechanism.
>
> A variant of DP, $\epsilon$-mutual-information differential privacy (MI-DP) [1], as mentioned by the `Reviewer wj8U`, bridges these two definitions. It is defined as follows:
>
> > **Definition**: ($\epsilon$-MI-DP).
> > According to [1], a mechanism $\mathcal{M}$ satisfies $\epsilon$-mutual-information differential privacy for some $\epsilon \in \mathbb{R}^{+}$ if, for *any neighboring inputs $X, X'$*, the conditional mutual information between $X$ and $Y=\mathcal{M}(X)$ conditioned on $X'$ satisfies
> > $$
> > I(X; Y | X') \leq \epsilon \text{ bits}.
> > $$
>
> > **Definition**: ($\epsilon$-MIP).
> > According to [2], a mechanism $\mathcal{M}$ satisfies $\epsilon$-MIP for some $\epsilon \in \mathbb{R}^{+}$ if, for *any input > $X$*, the mutual information between $X$ and the output $Y = \mathcal{M}(X)$ is bounded by $\epsilon$ bits, formally:
> > $$
> > I(X; Y) \leq \epsilon \text{ bits}.
> > $$
>
> The fundamental difference between MIP and DP, including its variant MI-DP, lies in their underlying **threat models**. DP operates under a strong adversary assumption, considering adversaries that have access to __neighboring input databases $X'$__. In contrast, MIP is designed under the assumption that the adversary has __no prior knowledge__ of the database, which aligns more closely with the threat model in our VFL setting.
>
> Given the assumption in our VFL setting where an adversary has *no* prior knowledge of the active party's data $\\{\mathbf{i}, \mathbf{y}, \mathbf{X}_0\\}$, MIP emerges as a more fitting choice. Our threat model, as elaborated in Section 3, coincides with the principles of MIP, making it a natural fit for our research.
>
> Meanwhile, [2] reveals a fundamental connection between mutual information and differential privacy bridged by *identifiability*. We have added the two references [1,2] in our updated version.
>
> --------
>
> **References**
>
> [1] Cuff Paul, and Lanqing Yu. "Differential privacy as a mutual information constraint." Proceedings of the 2016 ACM SIGSAC Conference on Computer and Communications Security. 2016.
>
> [2] Weina Wang, Lei Ying, and Junshan Zhang. "On the relation between identifiability, differential privacy, and mutual-information privacy." IEEE Transactions on Information Theory 62, no. 9 (2016): 5018-5029.